# POSITIVE DISTRIBUTION SHIFT AS A FRAMEWORK FOR UNDERSTANDING TRACTABLE LEARNING

## ABSTRACT

We study a setting where the goal is to learn a target function $f(x)$ with respect to a target distribution $D(x)$, but training is done on i.i.d. samples from a different training distribution $D'(x)$, labeled by the true target $f(x)$. Such a distribution shift (here in the form of covariate shift) is usually viewed negatively, as hurting or making learning harder, and the traditional distribution shift literature is mostly concerned with limiting or avoiding this negative effect. In contrast, we argue that with a well-chosen $D'(x)$, the shift can be positive and make learning easier – a perspective we call *Positive Distribution Shift (PDS)*. Such a perspective is central to contemporary machine learning, where much of the innovation is in finding good training distributions $D'(x)$, rather than changing the training algorithm. We further argue that the benefit is often computational rather than statistical, and that PDS allows computationally hard problems to become tractable even using standard gradient-based training. We formalize different variants of PDS, show how certain hard classes are easily learnable under PDS, and make connections with membership query learning.

## 1 INTRODUCTION

We consider a setting where our goal is to obtain good performance on some *target distribution* $\mathcal{D}$, but instead train on i.i.d. data from a different *training distribution* $\mathcal{D}'$. This type of distribution shift scenario is usually seen as less preferable than training on data from the target $\mathcal{D}$ itself. For example, we may want accurate pedestrian detection on images taken from cars in New York, but only have training data collected in California. Or we may wish to classify sentiment in scientific papers, yet train on Reddit posts. In practice, we rely on $\mathcal{D}'$ as a proxy because $\mathcal{D}$ is inaccessible, too costly to collect, or only available in very small amounts, while data from a different, but related, $\mathcal{D}'$ is easier to obtain. Indeed, most of the distribution shift literature is concerned with ensuring that training on $\mathcal{D}'$ is not (much) worse than training on $\mathcal{D}$.

The phenomenon that we study here is how training on an alternate training distribution $\mathcal{D}' \neq \mathcal{D}$ can be *beneficial*, and lead to improved learning compared to training *on the same number of samples* from $\mathcal{D}$. We are particularly interested in how this can be achieved without changing the training algorithm, but by simply running the "standard" training algorithm, such as Stochastic Gradient Descent (SGD) on some standard architecture. This matches current training practices, where much of the innovation and competitive advantage is not from new training algorithms nor new models, but rather from finding good training distributions $\mathcal{D}'$. In this paper, we set out to provide a concrete framework and theoretical foundation for studying this phenomenon.

We emphasize the *computational* benefit of the distribution shift, where the main benefit of training on an alternative $\mathcal{D}'$ is not in reducing the number of training examples required information theoretically, but in allowing training to be *tractable* (e.g. possible in polynomial time)[1]. We argue that this is the main benefit in practice, since sophisticated deep models are information theoretically learnable with sample complexity corresponding to the number of parameters, but also provably (worst case) computationally hard to learn. The real challenge in deep learning is therefore computational, and

---

[1]In this regard, our work differs from work on "helpful teachers" and the teaching dimension (Goldman et al., 1993; Goldman & Kearns, 1995; Zilles et al., 2011), which focus on reducing the number of examples required ignoring computational issues.

success rests on not being in this theoretical "worst case". Our goal is to show that with an appropriate distribution shift, tractable learning is possible, even with SGD.

The specific form of distribution shift that we consider here is *covariate shift* in supervised classification, where the goal is to learn a predictor $h : \mathcal{X} \to \mathcal{Y}$ under some target distribution $\mathcal{D}$ over $\mathcal{X}$, and we train using a different distribution $x \sim \mathcal{D}'$ but the same target conditional distribution $y|x$. We use $y|x = f(x)$ to denote this conditional distribution, where one can think of $f(x)$ as a random variable with a law determined by $x$.[2] We use $(x, y) \sim (\mathcal{D}, f)$ to denote the joint distribution over $(x, y)$ specified by $x \sim \mathcal{D}, y = f(x)$. We are therefore interested in a predictor $\hat{h} = A(S)$ obtained by running some (possibly randomized) learning rule $A$ on a training set $S \sim (\mathcal{D}', f)^m$ of $m$ i.i.d. samples from $\mathcal{D}'$. The performance is measured by the error $L_{\mathcal{D}, f}(\hat{h}) := \Pr_{(x,y) \sim (\mathcal{D}, f)} \left[ \hat{h}(x) \neq y \right]$, namely, the probability that $\hat{h}$ misclassifies an example drawn from the true target distribution $\mathcal{D}$. For a hypothesis class $\mathcal{H} \subseteq \mathcal{Y}^{\mathcal{X}}$, we denote by $L_{\mathcal{D}, f}(\mathcal{H}) = \inf_{h \in \mathcal{H}} L_{\mathcal{D}, f}(h)$ the minimal error achievable by a hypothesis in $\mathcal{H}$.

Our *Positive Distribution Shift (PDS)* learning framework is thus captured by the following definition.

**Definition 1.1** (PDS Learning)**.** A learning rule $A$ **PDS learns** $(\mathcal{D}, f)$ and $\mathcal{H}$ using the training distribution $\mathcal{D}'$, with sample size[3] $m(\epsilon)$ and runtime $T(\epsilon)$ if for every $\epsilon > 0$,

$$\mathbb{E}_{S' \sim (\mathcal{D}', f)^{m(\epsilon)}} \left[ L_{\mathcal{D}, f}(A(S')) \right] \leq L_{\mathcal{D}, f}(\mathcal{H}) + \epsilon$$

and $A(S')$ runs in time at most $T(\epsilon)$.

What Definition 1.1 is missing, and we will make concrete in different ways in subsequent Sections, are quantifiers over the learning rule $A$ and training distribution $\mathcal{D}'$, which are essential for discussing learning non-trivially.[4]

In this paper, we ask and formalize what it means to be able to learn a *hypothesis class* with "positive distribution shift", and give examples of what is, and is not, tractably learnable in this sense. To this end, we consider several hypothesis classes which are (information theoretically easy but) computationally hard to learn in a standard PAC setting (i.e., no poly-time learning algorithm can ensure learning with matching target and training distributions). Ultimately, we would like to ask, and rigorously answer, whether such classes are tractably learnable with positive distribution shift (under some concrete formalization), *by training a typical neural network using standard SGD*. Rigorously proving learnability using standard SGD on standard networks remains mostly elusive[5]. Towards this goal, for different classes, we give evidence in the form of (a) proving PDS learnability using *some* tractable algorithm (but not SGD on a network); (b) proving PDS learnability using a "stylized" SGD on some specific network; and/or (c) showing PDS learnability experimentally using standard SGD on a standard network.

**Realizability, label-noise, and agnostic learning setups.** In the *realizable* learning setting, the target function is deterministic and perfectly captured by the hypothesis class, that is, $f(x) = h^*(x) \in \mathcal{H}$ for some $h^* \in \mathcal{H}$, and $L_{\mathcal{D}, f}(\mathcal{H}) = 0$. In the *random classification noise* setting (with binary labels), we assume the existence of some $h^* \in \mathcal{H}$ such that for every input $x$, $\Pr[f(x) = h^*(x) \mid x] = 1 - \eta$, where the noise rate $\eta \in [0, \frac{1}{2})$ may be known or unknown. In this case, $L_{\mathcal{D}, f}(\mathcal{H}) = \eta$. In the general *agnostic* learning setting, we make no assumptions about the relationship between $f$ and the hypothesis class, and the goal is to compete with the best possible hypothesis in $\mathcal{H}$. Our Positive Distribution Shift learning framework applies to all of these settings, although we primarily focus on the label-noise setting.

---

[2] Formally, $f$ specifies a conditional distribution law for $y|x$, and for every observed $x_i$, the corresponding label $y_i$ is drawn independently from this conditional distribution–this is not a random function drawn once where $f(x_i)$ and $f(x_j)$ would then be dependent.

[3] For simplicity, we state learnability in expectation, however, all results hold with high probability, except for Theorem 4.5.

[4] Any single function $f$ is trivially and meaninglessly "learnable" with hard-wired $A$ that just always outputs $f$.

[5] Very few papers, if at all, truly analyze "standard" SGD, and even stylized SGD analysis is often technically complex.

## 2 WARM-UP: PARITIES ARE EFFICIENTLY LEARNABLE WITH PDS

To illuminate how positive distribution shift (PDS) can help tractability, we first consider the case of parity functions. A parity is a Boolean function $\chi_S : \{\pm 1\}^d \to \{\pm 1\}$ defined as $\chi_S(x) = \prod_{i \in S} x_i$, where $S \subseteq [d]$ is a subset of the coordinates of size $k \leq d$. Under uniform input distribution ($x \sim \text{Unif}\{\pm 1\}^d$), parities are *statistically* easy to learn: since there are $\binom{d}{k}$ candidate supports, then $O(k \log(d))$ random samples suffice to identify the true support with high probability. Even in the presence of label noise, the sample complexity remains $\text{poly}(d)$, thus distribution shift is not needed from an information-theoretical standpoint. *Computationally*, however, parities under the uniform distribution are believed to be hard. In particular, no efficient algorithm is known for learning noisy parities, and it is widely conjectured that even learning $\log(d)$-sparse parities requires super-polynomial time (Kearns, 1998; Blum et al., 2003; Applebaum et al., 2010; Feldman et al., 2006).

However, training on a different distribution $\mathcal{D}'$ can provide significant computational benefits, as it may reveal structure that is invisible under uniform inputs. For instance, consider a product distribution where the bits in the parity's support have non-zero bias. In this case, supported coordinates exhibit a nonzero correlation with the label, whereas unsupported coordinates remain uncorrelated. This additional structure and correlations (that is not visible on the uniform distribution) makes identifying the support tractably possible (even with SGD), which then makes the problem analogous to finding a linear predictor, which then generalizes back to the uniform distribution. In the terminology of Abbe et al. (2023a), this makes the target function be a staircase in the Fourier basis of $\mathcal{D}'$, so this positive distribution shift makes the function easy to learn. Either way, uniform inputs only reveal the relevant features through a $k$-way interaction, while the shift creates separate correlations for each relevant coordinate, turning an intractable problem into a tractable one.

However, when learning with gradient descent on standard architectures that are agnostic to the parity structure, training on a biased distribution alone is not sufficient, as our experiments show (Figure 1 (Right)). Malach et al. (2021) empirically demonstrated that when both the training and test distributions are taken to be $\mathcal{D}' = (1-p)\text{Uniform} + p\text{Biased}$ for some $p > 0$, noiseless parities of sparsity $\log(d)$ can be learned with a two-layer network of width $128$ trained with Adam on $d = 128$. They show that for $p > 0$, the network learns the parity with respect to $\mathcal{D}'$ (which includes the bias), whereas for $p = 0$ (i.e., $\mathcal{D}' = \mathcal{D} = \text{Uniform}$), it does not. Their focus is thus on PAC learning under the specific non-uniform distribution $\mathcal{D}'$. Our perspective differs: we aim to learn with respect to the uniform distribution $\mathcal{D}$ (or, more generally, any downstream test distribution), while only modifying the training distribution. In this setting, mixing biased and uniform samples is beneficial: the biased component reveals hidden correlations, while the uniform component ensures generalization to $\mathcal{D}$ and prevents overfitting to the biased distribution. Some of our analysis parallels that of Malach et al. (2021), but our view is different. Furthermore, we account for the presence of label noise. On the other hand, Cornacchia & Mossel (2023) show that training first on a randomly biased distribution $\mathcal{D}_1$ and then on $\mathcal{D} = \text{Uniform}$ enables learning parities under $\mathcal{D}$. Their emphasis is on curriculum learning as an instance of positive distribution shift with computational benefits. By contrast, we argue that the essential factor is not the order of training (curriculum) but the overall training distribution $\mathcal{D}'$.

In Section 4, we present our formal results on PDS learning of parities. First, in Theorem 4.3, we show that there is a simple tractable PDS learning algorithm for learning any parity function. Secondly, we show that parities are PDS learnable with analyzable (i.e. with layerwise training) gradient descent on a standard feed-forward neural network. Finally, we empirically show that even dense parities are PDS learnable with standard gradient descent on a two-layer feed-forward ReLU network. This extends both the results (Malach et al., 2021) and (Cornacchia et al., 2025), since their results only consider sparse parities. For a more comprehensive discussion of related work, see Appendix A.

## 3 ALL FUNCTIONS ARE EASY, WITH THE RIGHT TRAINING DISTRIBUTION

In Section 2, we saw that a computationally hard class, namely noisy parities, is easy to learn with PDS. A natural question to ask is: *what classes are tractably PDS learnable? Is it possible to PDS learn all functions representable by neural networks, i.e., all circuits?* Note that even constant-depth circuits are hard in the PAC framework

(Kharitonov, 1993; Daniely & Shalev-Shwartz, 2016; Daniely & Vardi, 2021). To answer these questions, we first introduce the most basic framework of positive distribution shift (PDS) learning, namely f-PDS learning. In this setting, the auxiliary distribution may depend not only on the test distribution $\mathcal{D}$ but also on the target function $f$ (hence the name function-dependent PDS). The training points are sampled from such a distribution $\mathcal{D}'$, and the learner's goal is to achieve low error on $\mathcal{D}$.

**Definition 3.1** (f-Dependent Positive Distribution Shift (f-PDS)). A hypothesis class $\mathcal{H}$ is f-PDS learnable with a learning algorithm $A$ with sample size $m(\varepsilon)$ and runtime $T(\varepsilon)$ if for every labeling rule $f : \mathcal{X} \to \Delta(\mathcal{Y})$ and every distribution $\mathcal{D}$ over $\mathcal{X}$, there exists an auxiliary distribution $\mathcal{D}'$ over $\mathcal{X}$ (allowed to depend on both $\mathcal{D}$ and $f$) such that for any $\varepsilon > 0$ with $m(\varepsilon)$ samples from $\mathcal{D}'$, $A$ has runtime $T(\varepsilon)$ and $\mathbb{E}_{S' \sim (\mathcal{D}', f)^{m(\varepsilon)}} \left[ L_{\mathcal{D},f}(A(S')) - L_{\mathcal{D},f}(\mathcal{H}) \right] \leq \varepsilon$.

It turns out that if allow the training distribution to depend on the target $f$, we can learn all poly-sized circuits with label noise. The algorithm is gradient descent on a non-standard neural network, namely, a network with non-standard topology or a fully connected feedforward network with non-standard activation functions and non-standard initialization. Essentially, we encode $f$ in $\mathcal{D}_f$ and the algorithm reconstructs the encoding from the samples.

*Boolean circuits*: Let $B$ be the set of gates $\{$AND, OR, NOT$\}$. A Boolean circuit $C$ on $d$ inputs is a finite directed acyclic graph (DAG) with input nodes $x_1, \ldots, x_d$, internal nodes labeled by gates in $B$, and one output node. For $x \in \{0,1\}^d$, values propagate along edges to define $f_C(x) \in \{0,1\}$. The size of the circuit is the number of gate nodes.

**Theorem 3.2** (Any Poly-sized Circuit with Label Noise is f-PDS Learnable with a Non-Standard Network). *For any $d, s \in \mathbb{N}$ there exists a fixed non-standard feed-forward neural network[6], an initialization, and a learning rate, such that any function $f : \{0,1\}^d \to \{0,1\}$ representable by (i) a circuit of size at most $s$, or (ii) a neural network of size at most $s$ with $\mathrm{polylog}(s)$ bits of precision, is f-PDS learnable with label noise by SGD on this neural network with samples and runtime $m(\epsilon), T(\epsilon) = \mathrm{poly}(d, s, \frac{1}{\epsilon})$.*

For the proof of Theorem 3.2 see Appendix B. Note that the sample complexity and the number of steps do not depend on the noise because we encode $f$ in $\mathcal{D}_f$. Encoding $f$ in $\mathcal{D}_f$ amounts to a form of "cheating", and the results rests on simulating PAC learning algorithms with (non-standard) GD on networks from (Abbe & Sandon, 2020). It shows that the notion of f-PDS might not be restrictive enough, i.e. it allows us to take an impossible choice for $\mathcal{D}_f$ that we couldn't be able to construct during training. It also makes the gradient descent variant considered here highly non-standard. Yet, Theorem 3.2 suggests the broader promise of f-PDS learning and naturally leads us to ask:

**Open Question 3.3** (Universality of f-PDS Learnability). *Is every function over $d$ inputs representable by a neural network of size $s$ (i.e. any computable function) f-PDS learnable by standard gradient descent on a neural network with standard[7] activation and initialization with runtime $\mathrm{poly}(d, s)$ and with $\mathrm{poly}(d, s)$ samples?*

This shifts the study about tractable learnability with neural networks to asking under what assumptions on the training distribution $\mathcal{D}'$ is the target PDS learnable.

## 4 A NEW LEARNING FRAMEWORK: DS-PAC

In this section, we prefer to not have the auxiliary distribution $\mathcal{D}'$ depend on the target $f$, but rather allow $\mathcal{D}'$ to depend only on the hypothesis class $\mathcal{H}$ and the target distribution $\mathcal{D}$. We introduce two variants of such positive distribution shift: *deterministic*, where the training set is sampled directly from the auxiliary distribution, and *randomized*, where we first draw a distribution from a meta-distribution (a distribution over distributions) and then sample the training set from it.

**Definition 4.1** (Deterministic Distribution-Shift PAC (D-DS-PAC)). A hypothesis class $\mathcal{H}$ is *Deterministic Distribution Shift PAC* (D-DS-PAC) learnable with a learning algorithm $A$ with sample size $m(\varepsilon)$ and runtime $T(\varepsilon)$ if for every distribution $\mathcal{D}$ over $\mathcal{X}$ and labeling rule $f : \mathcal{X} \to$

---

[6]the network is non-standard in either its topology (i.e. not fully connected) and activation with standard initialization or it is fully connected with standard activation function but with special initialization.

[7]Here, standard initialization refers to sampling from a fixed, reasonable distribution (e.g., Gaussian) with parameters independent of the target function $f$.

$\Delta(\mathcal{Y})$, there exists an auxiliary distribution $\mathcal{D}'$ over $\mathcal{X}$ (allowed to depend on $\mathcal{D}$ and $\mathcal{H}$ but not on $f$) such that for any $\varepsilon > 0$, $A$ has runtime $T(\varepsilon)$ on $m(\varepsilon)$ i.i.d. samples from $\mathcal{D}'$ and $\mathbb{E}_{S' \sim (\mathcal{D}',f)^{m(\varepsilon)}}\left[L_{\mathcal{D},f}(A(S')) - L_{\mathcal{D},f}(\mathcal{H})\right] \leq \varepsilon$. Formally, we say that a class $\mathcal{H}$ is D-DS-PAC learnable with label noise if this definition holds for a labeling rule that is induced by some $h^* \in \mathcal{H}$ with a label flipping noise $\eta \in [0, 1/2)$.

**Definition 4.2** (Randomized Distribution Shift PAC (R-DS-PAC)). A hypothesis class $\mathcal{H}$ is *Randomized Distribution-Shift PAC* (R-DS-PAC) learnable with a learning algorithm $A$ with sample size $m(\varepsilon)$ and runtime $T(\varepsilon)$ if for every distribution $\mathcal{D}$ over $\mathcal{X}$ and labeling rule $f : \mathcal{X} \to \Delta(\mathcal{Y})$, there exists a meta-distribution $\mathcal{M}_{\mathcal{D}}$ over distributions on $\mathcal{X}$ (allowed to depend on $\mathcal{D}$ and $\mathcal{H}$ but not on $f$) such that for any $\varepsilon > 0$, $A$ has runtime $T(\varepsilon)$ on $m(\varepsilon)$ i.i.d. samples from $\mathcal{D}' \sim \mathcal{M}_{\mathcal{D}}$ and $\mathbb{E}_{\mathcal{D}' \sim \mathcal{M}_{\mathcal{D}}} \mathbb{E}_{S' \sim (\mathcal{D}',f)^{m(\varepsilon)}}\left[L_{\mathcal{D},f}(A(S')) - L_{\mathcal{D},f}(\mathcal{H})\right] \leq \varepsilon$. Formally, we say that a class $\mathcal{H}$ is R-DS-PAC learnable with label noise if this definition holds for a labeling rule that is induced by some $h^* \in \mathcal{H}$ with a label flipping noise $\eta \in [0, 1/2)$.

In what follows, we study parities and juntas from the three perspectives on PDS learning: tractable, with analyzable GD, and empirical with standard GD.

**Parities.** For $S \subseteq \{1, \ldots, d\}$, define the parity $\chi_S : \{\pm 1\}^d \to \{\pm 1\}$ as $\chi_S(x) = \prod_{i \in S} x_i$, where $x = (x_1, \ldots, x_d) \in \{\pm 1\}^d$. The parity class over $d$ bits is defined as $\mathsf{Parity}_d = \{\chi_S : S \subset \{1, \ldots, d\}\}$. We consider also $k$-sparse parities, $\mathsf{Parity}_d^k = \{\chi_S : S \subset \{1, \ldots, d\}, |S| = k\}$. In this part, we establish the following: **(i)** noisy parities are D-DS-PAC learnable with a tractable algorithm (Theorem 4.3); **(ii)** noisy parities are D-DS-PAC learnable with a slightly stylized, analyzable variant of gradient descent (where the initialization depends on the sparsity of the parity; Theorem 4.5). Moreover, we show that noisy parities are f-PDS learnable using gradient descent on a feed-forward fully connected neural network for any test distribution, where for the uniform distribution such a result follows from Daniely & Malach (2020); **(iii)** empirically, standard gradient descent on a feed-forward neural network successfully learns noisy parities within the D-DS-PAC framework (Figure 1). This is in contrast to learning parity functions with noise in the PAC learning framework, which is believed to be computationally hard (Kearns, 1998; Blum et al., 2003; Applebaum et al., 2010; Feldman et al., 2006), both in general and even in the $\log(d)$-sparse case.

**Theorem 4.3** (Noisy Parities are Tractably D-DS-PAC Learnable). *There is an efficient D-DS-PAC learning algorithm $A$ that uses $\mathcal{D}' = \frac{1}{2}\mathrm{Rad}(1-\frac{1}{d})^{\otimes d} + \frac{1}{2}\mathrm{Unif}(\{\pm 1\}^d)$[8], such that for any distribution $\mathcal{D}$, the algorithm $A$ D-DS-PAC learns the class of parity functions $\mathsf{Parity}_d$ with label noise $\eta < \frac{1}{2}$ using $m(\epsilon) = O\left(\frac{d^2 \log \frac{2d}{\epsilon}}{(1-2\eta)^2}\right)$ samples from $\mathcal{D}'$ with runtime $T(\epsilon) = O\left(\frac{d^3 \log \frac{2d}{\epsilon}}{(1-2\eta)^2}\right)$.*

Next, we show that parities are D-DS-PAC learnable with a stylized analyzable stochastic gradient descent (SGD) on a 2-layer network, namely with layerwise training, $\ell_1$ regularization, and initializations that depend on the sparsity of the parity. The layerwise training assumption is standard in the theory of neural network learning literature (Abbe et al., 2023a; Barak et al., 2022; Bietti et al., 2022; Dandi et al., 2023), while the $\ell_1$ regularization and the dependence of initialization on sparsity are used to make the approach more analyzable. Our experiments use standard joint training of both layers and with standard initialization and confirm the theoretical results. Note that, in contrast to f-PDS, here the auxiliary distribution may depend only on the class of parities (and the sparsity) but not on the target function itself. For the proofs of Theorem 4.3 and Theorem 4.5, see Appendix C.

**Analyzable SGD 4.4** (Layerwise SGD with $\ell_1$-regularization). We perform stochastic gradient descent (SGD) on a two-layer network $f_{\mathsf{NN}}(x; \theta) = \sum_{j \in [N]} a_j \mathsf{ReLu}(\langle w_j, x \rangle + b_j)$, where $\theta = (a, W, b) \in \mathbb{R}^{N(d+2)}$ are the parameters of the network. We do layerwise training, where we first train the bottom layer while holding the top layer fixed and then train the top layer while holding the bottom layer fixed, with square loss and with $\ell_1$ regularization on the first layer with regularization parameter $\lambda$, and with stepsize $s$. We initialize hyperparameters as $a_j^0 \sim_{i.i.d.} \mathrm{Unif}(\{\pm 1\})$, $w_j^0 = 0$, and $b_j^0 \sim_{i.i.d.} \mathrm{Unif}(\{-d-1, -d, \ldots, d, d+1\})$ and take $N = (4d+6)$.

**Theorem 4.5** (Noisy Parities are D-DS-PAC Learnable with Stylized Analyzble GD). *For any fixed sparsity $k \leq d$, say $k = d/2$, of a parity function, the class $\mathsf{Parity}_d^k$ with label noise $\eta <$*

---

[8]A random variable has distribution $\mathrm{Rad}(p)$ if it takes the value $+1$ with probability $p$ and the value $-1$ with probability $1 - p$.

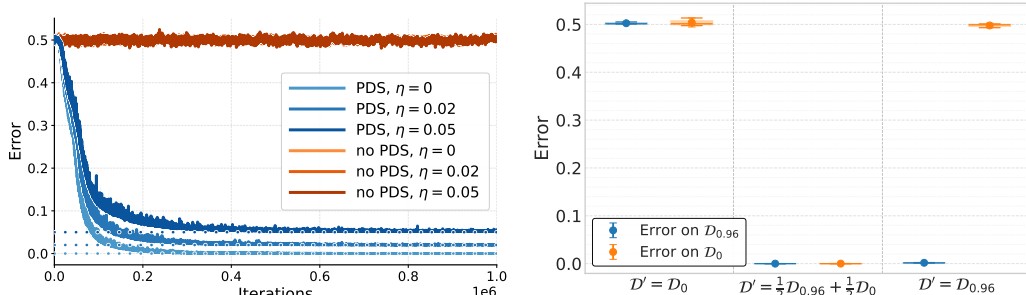

Figure 1: **Noisy Parities.** We study learning a degree-25 parity over 50 bits with label noise $\eta \in \{0, 0.02, 0.05\}$, using a two-layer ReLU network with 1024 hidden units trained by SGD (batch size 64, fresh samples, square loss, learning rate 0.01, both layers trained jointly). In the PDS setting, training samples are drawn from $\mathcal{D}' = \frac{1}{2}\mathcal{D}_{0.96} + \frac{1}{2}\mathcal{D}_0$, where $\mathcal{D}_\mu = \mathrm{Rad}((\mu + 1)/2)^{\otimes d}$ for $\mu \in [-1, 1]$; in the standard setting, from $\mathcal{D}' = \mathcal{D}_0$. The left panel shows test error on $\mathcal{D}_0$ during training, where PDS yields markedly more efficient learning; dotted lines indicate Bayes error ($\eta$). The right panel compares $\mathcal{D}' = \mathcal{D}_0$, $\mathcal{D}' = \frac{1}{2}\mathcal{D}_{0.96} + \frac{1}{2}\mathcal{D}_0$, and $\mathcal{D}' = \mathcal{D}_{0.96}$, plotting test error on $\mathcal{D}_{0.96}$ (blue) and $\mathcal{D}_0$ (orange) after $10^6$ steps. Only the mixture distribution achieves PDS generalization to the target $\mathcal{D}_0$.

$\frac{1}{2}$ *is D-DS-PAC learnable with Analyzable SGD 4.4 with $\lambda$ and $s$ that depend on $k$. For any distribution $\mathcal{D}$, if we train with $\mathcal{D}' = \frac{1}{2}\mathrm{Rad}(1 - \frac{1}{d})^{\otimes d} + \frac{1}{2}\mathcal{D}_2$, where $\mathcal{D}_2$ samples $l$ uniformly over $\{d, d-2, \ldots, -d\}$ then samples $\boldsymbol{x}$ uniformly conditioned on $\boldsymbol{1}^T\boldsymbol{x} = l$, using $m$ samples and $T$ steps of Analyzable SGD 4.4 with $m, T = O\left(\frac{d^9 \log(\frac{d}{\epsilon^2(1-2\eta)^2})}{\epsilon^3(1-2\eta)^6}\right)$, for any $k$ sparse noisy parity target $f$ we have $\mathbb{E}_{S' \sim (\mathcal{D}', f)^m}[L_{\mathcal{D}, f}(f_{NN}(x; \theta^{(T)}))] < \epsilon + \eta.$*

We empirically verify Theorem 4.5 using standard gradient descent on a two-layer fully connected ReLU network with standard initialization and training (Figure 1). For $\mu \in [-1, 1]$, let $\mathcal{D}_\mu$ be the distribution on $\{\pm 1\}^d$ with i.i.d. coordinates from $\mathrm{Rad}(\frac{\mu+1}{2})$, so that $\mathcal{D}_0 = \mathrm{Unif}\{\pm 1\}^d$. We study a degree-$k = 25$ parity over $d = 50$ bits with label noise, targeting the uniform distribution $\mathcal{D} = \mathcal{D}_0$. In the PDS setting, training samples come from $\mathcal{D}' = \frac{1}{2}\mathcal{D}_{0.96} + \frac{1}{2}\mathcal{D}_0$; in the standard setting from $\mathcal{D}$. Figure 1 (left) shows the evolution of the test error on $\mathcal{D}$ during training, where PDS yields significantly faster learning. The right panel compares training on $\mathcal{D}' = \mathcal{D}_0$ (left), $\mathcal{D}' = \frac{1}{2}\mathcal{D}_{0.96} + \frac{1}{2}\mathcal{D}_0$ (center), and $\mathcal{D}' = \mathcal{D}_{0.96}$, showing that only the mixture distribution (center), supports PDS generalization to the target $\mathcal{D}$.

*f-PDS learnability of parities.* f-PDS learnability of parities with analyzable GD with respect to the uniform test distribution follows form the existing result on learning parities with GD of Daniely & Malach (2020). Namely, they show that an analyzable version of gradient descent on 2-layer network learns parities with respect to the distribution $\mathcal{D}_1 = \frac{1}{2}\mathrm{unif}(\{\pm 1\}^d) + \frac{1}{2}\mathcal{D}_S$, where $\mathcal{D}_S$ is uniform over inputs outside the support of the parity, $S$, and on $S$ the inputs are all the same with probability $1/2$. This implies f-PDS learnability with respect to the uniform distribution (with training data still coming from $\mathcal{D}_1$, as needed in the definition of f-PDS). We further show that parities are *f-PDS learnable (i.e. with respect to any test distribution $\mathcal{D}$)* in Appendix C in Theorem C.4.

**Juntas.** For $k \leq d$, the $k$-junta class is $\mathsf{Junta}_d^k = \{f : \{0, 1\}^d \to \{0, 1\} : \exists S \subseteq [d], |S| \leq k, \exists g : \{0, 1\}^S \to \{0, 1\} \text{ s.t. } f(x) = g(x_S)\}$, where $x_S$ denotes the restriction of $x$ to the coordinates in set $S$. In this part, we establish the following: **(i)** noisy juntas are D-DS-PAC learnable by a Correlational Statistical Query (CSQ) algorithm (Theorem 4.6), **(ii)** noisy juntas are R-DS-PAC learnable with a stylized layerwise GD on a two-layer network with covariance loss, see Def. D.3 (Theorem 4.8), and **(iii)** empirically, noisy juntas are R-DS-PAC learnable with standard stochastic gradient descent with square loss on a feed-forward neural network (Figure 2). On the other hand, $\log(d)$-juntas are believed to be hard to learn in the PAC model, even in the realizable case (Applebaum et al., 2010; Chandrasekaran & Klivans, 2025).

Correlational Statistical Queries (CSQ) algorithms (Bendavid et al. (1995); Bshouty & Feldman (2002)) access the data via queries $\phi : \mathbb{R}^d \to [-1, 1]$ and return $\mathbb{E}_{x,y}[\phi(x)y]$ up to some error tolerance $\tau$. The algorithm in Theorem 4.6 is a CSQ algorithm.

**Theorem 4.6** (Noisy Juntas are D-DS-PAC Learnable). *There exists an input distribution $\mathcal{D}'$ and an algorithm $\mathcal{A}$ such that for every target distribution $\mathcal{D}$ and any $k$, the class of $k$-sparse juntas $\mathsf{Junta}_d^k$, with label noise $\eta < 1/2$, is D-DS-PAC learnable using $\mathcal{A}$ with $n = O(dk + 2^k)$ queries on $\mathcal{D}'$ of error tolerance $\tau = O((1 - 2\eta)2^{-k})$.*

We remark that a similar result can also be derived from Bshouty & Costa (2016) together with the connections between PDS and learning with membership queries established in Section 5 (see Appendix F for details). Our proof, however, shows D-DS-PAC learnability of noisy juntas directly by a different method, namely via correlation statistical queries. We also show that juntas are R-DS-PAC learnable with stylized analyzable GD, up to a multiplicative constant in the accuracy.

**Analyzable SGD 4.7** (Layerwise SGD with Covariance Loss). We perform gradient descent (SGD) on a two-layer network $f_{\mathsf{NN}}(\boldsymbol{x}; \boldsymbol{\theta}) = \sum_{j \in [N]} a_j \mathsf{ReLu}(\langle \boldsymbol{w}_j, \boldsymbol{x} \rangle + b_j)$, where $\boldsymbol{\theta} = (\boldsymbol{a}, \boldsymbol{W}, \boldsymbol{b}) \in \mathbb{R}^{N(d+2)}$ are the parameters of the network. We do layerwise training, where we first train the bottom layer while holding the top layer fixed and then train the top layer while holding the bottom layer fixed, with covariance loss. The covariance loss is defined for a sample $\{(x_i, y_i)\}_{i \in [m]}$ and a predictor $\hat{f}(x)$ as $L_{\text{cov}}(x, y, \hat{f}) = (1 - cy\bar{y})(1 - y\hat{f}(x))_+$, where $\bar{y} = \frac{1}{m} \sum_{i \in [m]} y_i$ and $c$ a positive constant such that $c \cdot \bar{y} < 1$. We initialize the first layer weights $w_{ij}^0 = 0$ for all $i \in [N], j \in [d]$, and the second layer weights $a_i^0 = \kappa$, for $i \in [N/2]$ and $a_i^0 = -\kappa$, for $i \in [N/2]$, where $\kappa > 0$ is a constant. The biases are initialized to $b_i^0 = \kappa$, for all $i \in [N]$, and after the first step of training, $b_i^1 \stackrel{i.i.d.}{\sim} \mathrm{Unif}[-L, L]$, where $L \geq \kappa$.

**Theorem 4.8** (Noisy Juntas are R-DS-PAC Learnable with Stylized Analyzable GD). *For any fixed $k \leq d$, the class of $k$-sparse juntas $\mathsf{Junta}_d^k$, with label noise $\eta < 1/2$, is R-DS-PAC learnable with respect to the uniform distribution $\mathcal{D}$ using Analyzable SGD 4.7. That is, for any $\epsilon > 0$ and noise level $\eta < 1/2$, there exists a meta distribution $\mathcal{M}_{\mathcal{D}'}$ such that after sampling $\mathcal{D}'$ from $\mathcal{M}_{\mathcal{D}'}$, Analyzable SGD 4.7 with batch size $B = \tilde{\Omega}(d \log(1/\epsilon))^9$ from $\mathcal{D}'$ on a two-layer network of width $\tilde{\Omega}(\epsilon^{-1}(1 - 2\eta)^{-1})$ after $T = \tilde{\Omega}(\epsilon^{-2}(1 - 2\eta)^{-2})$ steps of SGD learns any $k$ noisy junta target $f$ with error $\mathbb{E}_{\mathcal{M}_{\mathcal{D}}} \mathbb{E}_{S' \sim (\mathcal{D}', f)^m}[L_{(\mathcal{D}, f)}(f_{NN}(x; \theta^{(T)}))] < C_k(\eta + \epsilon^{c_k})$, for constants $c_k, C_k > 0$ that depend only on $k$.*

Theorem 4.8 builds on (Cornacchia et al., 2025, Theorem 5), extending it to include label noise and our PDS training distribution, which is not a product measure. It remains an open question whether D-DS-PAC learnability can be established using GD. The proofs of Theorem 4.6 and Theorem 4.8 are in Appendix D.

Finally, we empirically verify Theorem 4.8 holds for standard gradient descent training on a two-layer fully-connected ReLU network in Figure 2. For $\boldsymbol{\mu} \in [-1, 1]^{\otimes d}$, let $\mathcal{D}_{\boldsymbol{\mu}}$ be the distribution on $\{\pm 1\}^d$ with coordinates $x_i \sim \mathrm{Rad}((\mu_i + 1)/2)$, $i \in [d]$ (thus $\mathcal{D}_{\boldsymbol{0}} = \mathrm{Unif}\{\pm 1\}^d$). For $k \in \{7, 9\}$, we consider learning the following $k$-juntas: $f_k(x) := \prod_{i=1}^{k-2} x_i \cdot (1 + x_{k-1} + x_k - x_{k-1}x_k)$, with $x \in \{\pm 1\}^d$ and label noise, on the uniform target distribution (i.e. $\mathcal{D} = \mathcal{D}_{\boldsymbol{0}}$). Thus, $f_k$ is a $k$-junta with $\{\pm 1\}$ labels and minimum degree Fourier coefficient $k - 2$. In the PDS setting (R-DS-PAC), training uses samples from $\mathcal{D}' = \frac{1}{2}\mathcal{D}_{\boldsymbol{\mu}} + \frac{1}{2}\mathcal{D}_{\boldsymbol{0}}$, with $\boldsymbol{\mu} \sim \mathrm{Unif}[-1, 1]^{\otimes d}$, while in the standard setting (no PDS) it uses $\mathcal{D}$. In the left panel of Figure 2, we plot the evolution of the test error during training for learning $f_9$ on $d = 50$ bits, and observe that PDS yields far more efficient learning. In the right panel, we consider learning $f_7$ on varying input dimensions, and observe that the benefits of PDS increase with input dimension. We refer to Appendix G for the experiment's details and additional plots.

We would like to explore PDS learning for additional well-studied hypothesis classes. We will now do so by connecting PDS learning to learning with membership queries.

---

[9]Here, for $c \in \mathbb{R}$, $\tilde{\Omega}(d^c) = \Omega(d^c \mathrm{poly} \log(d))$.

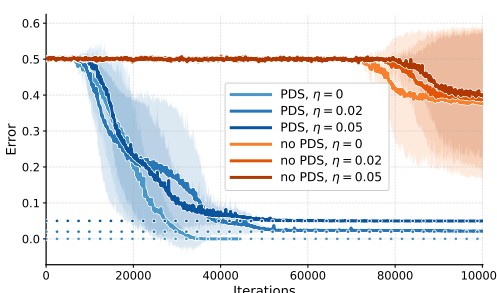 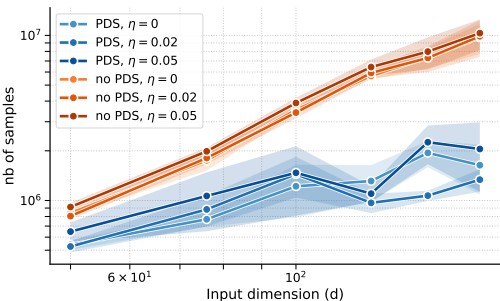

Figure 2: **Noisy juntas.** (Left) We consider learning $f_9$ (see Sec. 4) over $d = 50$ bits on $\mathcal{D} = \mathrm{Unif}\{\pm 1\}^d$ with a two-layer ReLU network (1024 hidden units) trained with SGD (batch size 64, fresh samples, square loss, l.r. 0.01, both layers trained jointly). In the PDS setting we train on $\mathcal{D}' = \frac{1}{2}\mathcal{D}_{\boldsymbol{\mu}} + \frac{1}{2}\mathcal{D}_{\mathbf{0}}$, with $\boldsymbol{\mu} \sim \mathrm{Unif}[-1,1]^{\otimes d}$ (and where $\mathcal{D}_{\boldsymbol{\mu}} := \otimes_{i \in [d]} \mathrm{Rad}((\mu_i + 1)/2)$), while in the standard (no PDS) setting $\mathcal{D}' = \mathcal{D}$. We plot the test error on $\mathcal{D}$ during training; the dotted lines show the Bayes error (i.e. $\eta$). (Right) We consider learning $f_7$ on $\mathcal{D} = \mathrm{Unif}\{\pm 1\}^d$ and we plot the sample complexity needed to reach within 0.01 of Bayes error versus the input dimension. In both plots, PDS training is markedly more efficient.

## 5 DS-PAC AND MEMBERSHIP QUERIES

In this section, we connect our PDS framework to the classical model of learning with membership queries (MQ). In this model, the learning algorithm can actively query specific points and obtain their labels from the target function $f$, in addition to sampling random labeled examples from the target distribution. The MQ model is strictly stronger than PAC (see Appendix A). Importantly, queries can be issued *adaptively*, that is, based on previously obtained labels, making it strictly more powerful than our "passive" PDS model, where examples are drawn non-adaptively from an auxiliary distribution.

The non-adaptive setting, defined below, has also been studied for specific concept classes (in the realizable case), including parities and DNFs under the uniform distribution (Feldman, 2007), juntas (Bshouty & Costa, 2016), and decision trees of logarithmic depth in the input dimension (Bshouty, 2018).

**Definition 5.1** (Non-Adaptive Membership Queries (NA-MQ)). A hypothesis class $\mathcal{H}$ over domain $\mathcal{X}$ is NA-MQ learnable with sample size $m(\varepsilon)$ if there exist integers $m_{\mathrm{rand}}(\varepsilon)$ and $m_{\mathrm{mq}}(\varepsilon)$ with $m_{\mathrm{rand}}(\varepsilon) + m_{\mathrm{mq}}(\varepsilon) = m(\varepsilon)$, a distribution $\mathcal{Q}$ over batches $\tilde{S}_{\mathcal{X}} \in \mathcal{X}^{m_{\mathrm{mq}}(\varepsilon)}$, and a learning algorithm $A$, such that the following holds. For every distribution $\mathcal{D}$ over $\mathcal{X}$ and every labeling rule $f : \mathcal{X} \to \Delta(\mathcal{Y})$, the distribution $\mathcal{Q}$ may depend on $\mathcal{D}$ and on known complexity parameters of $\mathcal{H}$, but not on the unknown target $f$ and not on any observed labels. The learner has access to an i.i.d. labeled sample $S_{\mathrm{rand}} = \{(x, f(x)) : x \sim \mathcal{D}\}$ of size $m_{\mathrm{rand}}(\varepsilon)$, and to a non-adaptive query batch $\tilde{S}_{\mathcal{X}} \sim \mathcal{Q}$ of size $m_{\mathrm{mq}}(\varepsilon)$ with labels $S_{\mathrm{mq}} = \{(x, f(x)) : x \in \tilde{S}_{\mathcal{X}}\}$. The algorithm outputs $A(S_{\mathrm{rand}}, S_{\mathrm{mq}})$ such that $\mathbb{E}_{S_{\mathrm{rand}} \sim (\mathcal{D},f)^{m_{\mathrm{rand}}(\varepsilon)}, \tilde{S}_{\mathcal{X}} \sim \mathcal{Q}}\left[ L_{\mathcal{D},f}(A(S_{\mathrm{rand}}, S_{\mathrm{mq}})) - L_{\mathcal{D},f}(\mathcal{H}) \right] \le \varepsilon$. We say the learner is *deterministic NA-MQ* if $\mathcal{Q}$ is a point mass (the query set is fixed), and *randomized NA-MQ* otherwise.

Note that random classification noise is defined in the same way as in the PAC model, namely by flipping labels returned by $f$. As we show below, this non-adaptive model plays a central role in the DS-PAC framework. The proof is in Appendix E.

**Theorem 5.2** (D-DS-PAC $\to$ NA-MQ). *For any hypothesis class $\mathcal{H}$, every D-DS-PAC-learning algorithm for $\mathcal{H}$ under label noise yields a NA-MQ-learning algorithm for $\mathcal{H}$ under label noise with the same sample complexity and running time.*

**Theorem 5.3** (NA-MQ $\to$ R-DS-PAC). *Fix a hypothesis class $\mathcal{H}$ and a distribution $\mathcal{D}$ over $\mathcal{X}$. Suppose there is a (possibly randomized) non-adaptive MQ learner $A$ for $\mathcal{H}$ under $\mathcal{D}$ that, using at most $m_0(\varepsilon)$ queries, outputs $\hat{h}$ with $\mathbb{E}\left[L_{\mathcal{D},f}(\hat{h})\right] \le L_{\mathcal{D},f}(\mathcal{H}) + \varepsilon$. Then $\mathcal{H}$ is R-DS-PAC-learnable under $\mathcal{D}$ with sample size $m(\varepsilon) = O\left(m_0(\varepsilon/2)\left(\log m_0(\varepsilon/2) + \log(1/\varepsilon)\right)\right)$. Moreover, if $A$ is deterministic non-adaptive, then $\mathcal{H}$ is D-DS-PAC-learnable under $\mathcal{D}$ with the same sample bound.*

Moreover, we can show that a deterministic NA-MQ algorithm implies D-DS-PAC. As an immediate application, we obtain that DNFs under the uniform distribution and decision trees of logarithmic depth are R-DS-PAC learnable.

Natural open problems include the following: **(i)** Can DNFs and decision trees be learned with PDS by training a neural network with gradient descent? **(ii)** DNFs under the uniform distribution and log-depth decision trees are learnable in NA-MQ by identifying heavy Fourier coefficients. A natural broader question is whether the class of sparse functions, those with only a few non-zero Fourier coefficients, is also NA-MQ learnable under the uniform distribution using a number of queries polynomial in the dimension and the sparsity parameter, and hence R-DSPAC learnable. This is known to hold in the adaptive MQ model (Mansour, 1994). **(iii)** Can we separate learning with adaptive MQ from NA-MQ/R-DS-PAC for natural classes? We note that Feldman (2007) sketches such a separation for certain artificial classes. For functions representable by polynomial-size circuits, we know that they are not R-DS-PAC learnable because constant-depth circuits are known to be hard to learn even with adaptive membership queries (Kharitonov, 1993).

## 6 SUMMARY AND OPEN QUESTIONS

Much of the effort in the current practice of ML is in "dataset selection" or finding the best training distribution $\mathcal{D}'$ in order to get good results on a target $\mathcal{D}$. It is important to develop a framework, language, methodology and theory to capture this. This paper is our attempt to make progress toward this goal.

In Section 3 on f-PDS, we asked which targets are learnable using some training $\mathcal{D}'$, with the f-PDS framework. The definition allows $\mathcal{D}'$ to depend on $f$, so this is not a "recipe" where one can construct $\mathcal{D}'_f$ during training. In particular, our Theorem 3.2 shows the existence of $\mathcal{D}'_f$ by "cheating" and leaking information through an unnatural and impossible choice of $\mathcal{D}'_f$, thus not providing insight but rather helping us refine the question to ask. This is done in Open Question 3.3 where we limit training to a fixed training rule that does not process "leaked" information: SGD on a regular neural net—the main contribution of this Section is thus in formulating Open Question 3.3. The insight is two-fold: (a) by understanding *which* $\mathcal{D}'_f$ allows learning $(\mathcal{D}, f)$, we can gain insight on the type of properties of "good" training distributions $\mathcal{D}'$, and hopefully guidance into the "dataset selection" problem; (b) shift the study about tractable learnability of neural networks from asking "which subset of functions representable by neural networks are tractably learnable" (e.g. in Daniely & Vardi (2020); Daniely et al. (2023)), or even "under what simple, e.g. uniform/Gaussian/etc input distribution" (e.g. as in Chen et al. (2022); Daniely & Vardi (2021)), to asking, "for any function $f$ representable by a neural network and input $\mathcal{D}$, under what assumptions on the training distribution $\mathcal{D}'$ is it PDS learnable?".

In Sections 4 and 5 on DS-PAC and membership queries, we turned to a more prescriptive order of quantifiers, where here the training distribution $\mathcal{D}'$ depends only on the hypothesis class $\mathcal{H}$ and input distribution $\mathcal{D}$, but not on the target $f$. We present concrete definitions of a notion of learnability that we advocate studying, and initial results showing its power and depth. In particular, we show strong connections with membership query learning, and especially *non adaptive* membership query learning. NA-MQ has only been sporadically studied in the past, and frequently not explicitly but by providing MQ methods that do not require adaptation. We hope this will reignite interest in NA-MQ, and more directly in DS-PAC learning.

If we allow an arbitrary tractable algorithm and to randomly choose a training distribution $\mathcal{D}'$, then DS-PAC is "equivalent" (with a small increase in sample complexity) to NA-MQ. With DS-PAC, unlike NA-MQ, we are mainly interested in using a particular learning rule, namely SGD on a neural network. Our work raises the following **Open Question**: *are interesting hypothesis classes that are tractably learnable with NA-MQ (and thus also R-DS-PAC) also DS-PAC learnable using SGD on a neural network? E.g., decision trees?* An interesting candidate class of particular interest is the class of functions with a sparse Fourier decomposition. **Open Question**: *Are sparse functions DS-PAC learnable? What about with using SGD on a neural network?* On the negative side, it would be good to show hardness of DS-PAC learning, which is equivalent to hardness of NA-MQ learning. **Open Question**: *Are there natural classes that are MQ learnable, but not NA-MQ/DS-PAC learnable?* We are not aware of any work specifically on showing adaptivity is necessary for MQ, i.e. seperating A-MQ and NA-MQ for natural classes. Another open question concerns the necessity of randomization in R-DS-PAC. That is, whether there are classes that are R-DS-PAC learnable but not

D-DS-PAC, and whether this gap exists specifically for SGD on a neural network. In particular, are juntas D-DS-PAC learnable with SGD on a neural network? Overall, we hope our work will create interest in PDS learning, which we view as an important learning paradigm. We can summarize the PDS hierarchy implied by our work as

$$\text{PAC} \rightleftarrows \text{D-DS-PAC} \rightarrow \text{R-DS-PAC} \leftrightarrow \text{NA-MQ} \rightleftarrows \text{A-MQ}.$$

Table 1 provides a summary of the current classification of hypothesis classes in the PDS hierarchy.

| Class | Tractable Algorithm | "Analyzable" GD | Experiment (realistic GD) |
|---|---|---|---|
| Parity | D-DS-PAC | D-DS-PAC$^{\text{GD}}$ | Figure 1 |
| | Theorem 4.3 | Theorem 4.5 | |
| $k$-Juntas | D-DS-PAC | R-DS-PAC$^{\text{GD}}$ | Figure 2 |
| | Theorem 4.6 | Theorem 4.8 | |
| $\log(d)$-depth Decision Trees | R-DS-PAC/NA-MQ | — | — |
| | Bshouty (2018) | | |
| DNF (uniform dist.) | R-DS-PAC/NA-MQ | — | — |
| | Feldman (2007) | | |
| Poly-Sized Circuits | f-PDS | f-PDS (with special | — |
| | Theorem 3.2 | unrealistic network) | |
| | (not R-DS-PAC) | Theorem 3.2 | |

Table 1: We study the learnability of concept classes under random classification noise across multiple frameworks. These classes are either known to be hard to learn under well-established cryptographic assumptions or are widely believed to be hard. We ask whether they can instead be learned efficiently under positive distribution shift (allowing any algorithm), via provably analyzable gradient descent on neural networks, and empirically using standard gradient descent. For each concept class, the test distribution is arbitrary, except for DNFs, where it is restricted to the uniform distribution. Notably, Theorem 4.8 was also established implicitly by Cornacchia et al. (2025).

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

# A    ADDITIONAL RELATED WORK

**GD on neural networks.**   A large body of work has analyzed the sample and time complexity of learning simple function classes—such as parities, juntas, and single/multi-index models—using stylized gradient descent on shallow networks under standard symmetric input distributions (e.g., uniform Boolean, standard Gaussian) (Ben Arous et al., 2021; Bietti et al., 2022; Abbe et al., 2023a; Dandi et al., 2023; 2024; Lee et al., 2024; Troiani et al., 2024; Arnaboldi et al., 2024; Şimşek et al., 2024; Damian et al., 2025; Joshi et al., 2025). For parities, it is known that under uniform Boolean inputs they are learnable by Stochastic Gradient Descent (SGD) with small batch size on an emulation network (Abbe & Sandon, 2020), but require at least $\Omega(\binom{d}{k})$ time in the Statistical Query (SQ) model (Kearns, 1998) or with GD under limited gradient precision (Abbe & Sandon, 2020). Positive results for gradient descent on standard shallow networks exist for sparse parities $(k = O_d(1))$, matching the SQ lower bound (Barak et al., 2022; Glasgow, 2023; Kou et al., 2024), and for dense parities depending on the initialization (Abbe & Boix-Adsera, 2022; Abbe et al., 2024). For juntas under uniform Boolean inputs, complexity is governed by the Fourier-Walsh structure of the target, captured by the leap for square loss (Abbe et al., 2022a;b; 2023a) and the SQ-leap for general losses (Joshi et al., 2024). Our work instead seeks to overcome these barriers via Positive Distribution Shift (PDS).

Beyond symmetric inputs, several works study structured data distributions—e.g., spike-covariance (Mousavi-Hosseini et al., 2023), hierarchical (Mossel, 2016; Cagnetta et al., 2024; Cagnetta & Wyart, 2024; Dandi et al., 2025), or non-centered product measures (Malach et al., 2021; Cornacchia et al., 2025)—showing that structure can reduce sample complexity relative to unstructured settings. In particular, Malach et al. (2021) analyze sparse parities under a mixture of uniform and biased distributions, but only for a differentiable model combining a linear predictor with an appended parity module, and with no label noise. By contrast, we consider standard two-layer architectures and demonstrate PDS benefits also for high-degree parities, and with label noise. For juntas, Cornacchia et al. (2025) show gains from training on randomly shifted inputs, but do not examine whether such favorable distributions can transfer to other target distributions, such as the uniform.

Other works have studied distribution shift between train and test, as we do here. A prominent line of research analyzes models tested outside their training domain (e.g., out-of-distribution, length generalization (Anil et al., 2022; Abbe et al., 2023b; Zhou et al., 2023; Abbe et al., 2022b; Power et al., 2022)), probing whether neural networks rely on reasoning or memorization. Transfer learning has also been examined as a way to cope with scarce target data (Damian et al., 2022; Gerace et al., 2022; Ingrosso et al., 2025). These works often treat distribution shift negatively—either as a stress test of extrapolation or as a necessity when labeled target data is limited. In contrast, we adopt a positive perspective, developing a unifying theory showing how a carefully chosen training distribution can actively facilitate learning. Closer to our approach, curriculum learning studies investigate training on a *sequence* of distributions of increasing complexity, and demonstrate benefits of ordering (Saglietti et al., 2022; Cornacchia & Mossel, 2023; Abbe et al., 2023c; Mannelli et al., 2024; Mignacco & Mori, 2025; Wang et al., 2025). By contrast, we focus on a *single* positively shifted training distribution and its ability to transfer to the target distribution, arguing that one well-chosen source can substantially ease learning.

**Smoothed analysis.**   Smoothed analysis (Spielman & Teng, 2004) blends worst-case and average-case perspectives by measuring the maximum expected performance of an algorithm under slight random perturbations of its inputs. Mossel et al. (2004) showed that such perturbations alter the Fourier coefficients of Boolean functions and can drastically reduce complexity. Subsequent works established smoothed-analysis guarantees for learning juntas and decision trees under random product distributions (Kalai & Teng, 2008), and for DNFs (Kalai et al., 2009). The ID3 decision-tree algorithm was later shown to efficiently learn juntas in the smoothed model (Brutzkus et al., 2020), and these ideas were extended to general Markov Random Fields with a smoothed external field (Chandrasekaran & Klivans, 2025). Related work also studied smoothing via additive Gaussian noise in the Gaussian input setting (Klivans & Meka, 2013). Our work departs from this literature in two key ways: 1) Positive Distribution Shift (PDS) involves deliberate, often large and typically deterministic shifts (except in the case of R-DS-PAC), rather than small random perturbations; and 2) we focus on providing quantitative computational guarantees for concrete algorithms, including gradient descent on neural networks.

**Learning with (adaptive) membership queries.** Classical results in this model established its power relative to the standard PAC framework. Angluin's foundational work showed that finite automata can be exactly learned from queries and counterexamples Angluin (1987), and that monotone DNF formulas are efficiently learnable with membership queries Angluin (1988). Subsequent work extended these results to richer classes: Bshouty (1993) gave polynomial-time algorithms for decision trees, while Schapire & Sellie (1993) developed methods for learning sparse multivariate polynomials over GF(2). Fourier-analytic techniques played a central role in later breakthroughs: Linial et al. (1993) showed that constant-depth circuits (AC$^0$) can be efficiently learned from membership queries using low-degree Fourier concentration, and Kushilevitz & Mansour (1993) applied similar ideas to decision trees under the uniform distribution. Building on this line, Jackson (1997) obtained an algorithm for learning DNF formulas with membership queries under the uniform distribution.

**Learning from random walks.** The passive learning model via random walks (Aldous & Vazirani, 1995; Bartlett et al., 1994; Gamarnik, 1999; Roch, 2007) lies strictly between uniform-distribution learning and the membership query model: it is weaker than the latter, yet stronger than the former. This model has enabled several important results, including learning DNFs and decision trees with random classification noise under the uniform distribution (Bshouty et al., 2005), learning parities with noise (as a consequence of (Bshouty & Feldman, 2002; Bshouty et al., 2005)), and agnostic learning of juntas (Arpe & Mossel, 2008). Similar to the membership query setting, random classification noise can be tolerated whenever there exists an algorithm for the realizable case. In the context of this paper, the random walk model can be viewed as a specific case of R-DS-PAC.

# B  PROOFS FOR F-PDS

*Proof of Theorem 3.2.* If we have freedom to change the network architecture, activation, initialization, and learning rate, we can use the PAC-universality of neural networks to show Theorem 3.2.

Note that it suffices to show that we can transmit $\text{poly}(d)$ bits by encoding them into a distribution $\tilde{\mathcal{D}}$. We can use this to decode the network using a PAC learning algorithm. Simulating this decoding algorithm with GD on the non-standard network gives us the desired learning algorithm.

We first show the subroutine of this PAC algorithm for transmitting bits using a distribution.

**Lemma B.1** (Transmitting $r$ bits using a distribution). *Let $\theta$ be a set of $r = \text{poly}(d) < 2^{d-1}$ elements from $\{0,1\}^d$. There exists a $\text{poly}(m,d)$ procedure **DECODE** such that for any $d$ and any $r \leq \text{poly}(d)$ there are a distribution $\mathcal{D}$ over $\{0,1\}^d$ and $m \leq \text{poly}(r,d,\log\frac{1}{\epsilon})$ such that for $S = \{x_i\}_{i=1}^m \sim (\mathcal{D},f)^m$ and $\hat{\theta} = \textbf{DECODE}(S)$ we have $\hat{\theta} = \theta$ with probability at least $1 - \epsilon$.*

*Proof of Lemma B.1.* Associate each $x \in \{0,1\}^d$ with an integer $0, 1, \ldots, 2^d - 1$. Define $\mathcal{D}$ as follows: for $i = 1, \ldots, r$, $P(x = i) = \begin{cases} \frac{2}{5r}, & \text{if } \theta[i] = 0 \\ \frac{3}{5r}, & \text{if } \theta[i] = 1. \end{cases}$ . For $i > r$, $P(x = i) = 0$, and $P(x = 0) = 1 - \sum_{i \neq 0} P(x = i)$. We will need $O\left(\frac{r(\log r + \log\frac{1}{\epsilon})}{(\frac{1}{r})^2}\right) = O\left(r^3(\log r + \log\frac{1}{\epsilon})\right)$ samples from $\mathcal{D}$. Let $\hat{P}$ be the empirical distribution we get on $\{0,1\}^d$ from the sample $S = (x_1, \ldots, x_m)$. With this many samples, we have that with probability $\geq 0.99$, we have $\hat{P}(x = i) \in (P(x = i) - \frac{1}{20r}, P(x = i) + \frac{1}{20r})$, so we can set the threshold $\hat{P}(x = i) = \frac{2.5}{5r}$ to recover the bit. The runtime of this procedure is $O(mrd) = \text{poly}(r, \log\frac{1}{\epsilon})$. To see why this many samples are sufficient, let $Y_i = \sum_{j=1}^m \frac{\mathbf{1}_{x_j = i}}{m}$. Note that $\mathbb{E}(\mathbf{1}_{x_i = i}) = P(x = i)$, so by Hoeffding inequality applied to $Y_i - P(x = i)$ with $0 \leq \frac{\mathbf{1}_{x_j = i}}{m} \leq \frac{1}{m}$

$$P\left(\cap_{i=1}^r |Y_i - P(x = i)| < \frac{1}{20r}\right) \geq 1 - 2r\exp\left(-\frac{\frac{1}{20^2 r^2}}{m\frac{1}{m^2}}\right) = 1 - 2r\exp(-\frac{m}{20^2 r^2}).$$

Therefore, if we recover bits using $\hat{\theta}_i = \begin{cases} 1 & \text{if } Y_i \geq \frac{3}{5r} \\ 0 & \text{if } Y_i < \frac{3}{5r} \end{cases}$, then with the same probability as above we recover the bits correctly, $P(\hat{\theta} = \theta) \geq 1 - 2r\exp(-\frac{m}{20^2 r^2})$. Taking $m = O(r^2(\log r + \log\frac{1}{\epsilon}))$ suffices to have $2r\exp(-\frac{m}{20^2 r^2}) \leq \epsilon$, i.e. to transmit the bits with probability $\geq 1 - \epsilon$. $\square$

Consider learning without noise first. We will add the label noise later.

**Encoding the network.** Let $f : \{0,1\}^d \to \{0,1\}$ be boolean circuit satisfying of poly size. Then $f$ can be encoded in $r = \text{poly}(d)$ atoms of $\{0,1\}^d$.

For the circuits (i) we use the following encoding: we encode the adjacency matrix of size $M \times M$ of the DAG representing the boolean circuit, where $M = \text{poly}(d)$ is the total number of nodes of the DAG, with the value at entry $(i,i)$ states whether the node is input, output, or which of the gate node types, and the value at $(i,j)$ entry represents if there is an edge and if yes what direction the edge is between node $i$ and node $j$. Note that from the adjacency matrix we can reconstruct the function in time $\text{poly}(d)$. So, for elements $(x^1, \dots, x^d) \in \{0,1\}^d$ we encode

- Use the first $\log \text{poly}(d)$ bits to encode the the size of the circuit, $M$, and the total number of bits needed to transmit, $r$

- Use the next $2 \log \text{poly}(d)$ bits to encode the value of the coordinates $i$ and $j$ of the entry $(i,j)$ in the adjacency matrix.

- Use the next $4$ bits to encode the value of the entry.

- Use the remaining bits to mark that the atom is coming from this distribution by setting them to 1, note that there is at least $cn$ of these for some $c$ (that can be taken to be $\frac{1}{4}$ for large enough $n$).

For poly-sized networks we use the following encoding: We can do the encoding similarly: (i) Use the first $3 \log \text{poly}(d)$ bits to encode the depth $L$, width $W$, and total number of bits $r$, (ii) Use the next $3 \log \text{poly}(d)$ bits to location of the weight in the format $(i,j)$, which represents depth and the width of that layer in which the weight is located, (iii) use the next $d/2$ bits encode the weight value (e.g. if we want rational weights, we can use the first $d/4$ bits for the numerator and the next $d/4$ bits for the denominator, (iv) use the remaining bits to mark that the atom is coming from this distribution by setting them to 1, note that there is at least $cn$ of these for some $c$ (that can be taken to be $\frac{1}{4}$ for large enough $n$).

Note that the width and the depth are encoded in all $x \in \theta$ (and if the whole layer is missing, we can tell from other points). So, similarly, in the case that we do not recover $\theta$, we can infer that we didn't recover it by counting the number of distinct weight positions recovered. In this case, we will return the zero predictor.

The rest of the proof is analogous for circuits and networks.

Since there is $\text{poly}(d)$ gates, this encoding procedure requires $r = \text{poly}(d)$ elements $x_i \in \{0,1\}^d$.

**PAC Learning Algorithm For f-PDS.** Call the set of $r$ elements $x_i \in \{0,1\}^n$ that encode the circuit $\theta$. Now using Lemma B.1, there is a distribution $\mathcal{D}_\theta$ and a PAC learning algorithm $\mathcal{A}$ which using $m = O\left(r^2(\log r + \log \frac{1}{\varepsilon})\right)$ samples $S$ from $S \sim \mathcal{D}_\theta^m$ recovers $\mathcal{A}(S) = \hat{\theta}$ such that $\hat{\theta} = \theta$ with probability at least $1 - \epsilon$.

With probability $\epsilon$ it happens that we do not recover all of $\theta$. Note that by the construction of $\mathcal{D}_\theta$ from Lemma B.1, any $x \sim \mathcal{D}_\theta$ from the support of this distribution is in $\theta$. Note that the size of the adjacency matrix is encoded in any of the elements in the support of $\mathcal{D}_\theta$, so we can tell if do not recover $\theta$. In this case, we will return the zero predictor.

This gives a PAC learning algorithm $\mathcal{A}'$ to recover $f$ using $m = O\left(r^2(\log r + \log \frac{1}{\varepsilon})\right)$ samples from $\mathcal{D}_\theta$ as described above, $\mathcal{A}'(S) = \begin{cases} \mathcal{A}(S) & \text{if we recover all of } \theta \\ \hat{f} \equiv 0 & \text{if we don't recover } \theta. \end{cases}$

Note that with $m' := 4m + \log(1/\epsilon) = O\left(r^2(\log r + \log \frac{1}{\varepsilon})\right)$ samples from $\mathcal{D}_f := \frac{1}{2}\mathcal{D}_\theta + \frac{1}{2}\mathcal{D}$ the algorithm $\mathcal{A}'$ has the same output with probability at least $1 - 2\epsilon$, i.e. recovers $f$, and with probability $2\epsilon$ returns 0. This implies that the square loss of $\mathcal{A}'$ over any test distribution is at most $2\epsilon$, so the square loss over $\mathcal{D}_f$ is at most $2\epsilon$ as well. We can rename here $\epsilon = \frac{\epsilon}{2}$.

Adding the noise back, note that $\mathcal{A}'$ does not use the labels, so the same proof shows that with $m = O\left(r^2(\log r + \log \frac{1}{\varepsilon})\right)$ samples from $\mathcal{D}_f$, $\mathcal{A}'$ learns $f$ exactly with probability $1 - \epsilon$ and with probability $\epsilon$ outputs 0.

So with noise, $\mathcal{A}'$ achieves population risk of at most $\mathbb{E}_{S \sim (\mathcal{D}_f, f)^m}[L_{(\mathcal{D},f)}(\mathcal{A}'(S))] \leq \epsilon(1 - \eta) + (1 - \epsilon)\eta \leq \epsilon + \eta$. Furthermore, with the same number of samples $\mathcal{A}'$ achieves expected square loss of at most $\epsilon + \eta$.

Note that $\mathcal{A}'$ is a tractable PAC learning algorithm for learning $f$ on $\mathcal{D}_f$. More imporantly, $\mathcal{A}'$ is an f-PDS learning algorithm for learning $f$ on $\mathcal{D}$ with samples from $\mathcal{D}_f$, since $\mathcal{A}'$ exactly learns $f$ with probability $1 - \epsilon$ and returns 0 otherwise.

We summarize the f-PDS result in the following lemma and then focus on simulating this with GD on a non-standard network.

**Lemma B.2** (f-PDS Tractable Algorithm For Learning Any Circuit). *Let $f : \{0,1\}^d \to \{0,1\}$ be a function that satisfies either (i) $f$ is a circuit of any depth and of size $s = \text{poly}(d)$ OR (ii) $f$ is a fully connected feed forward ReLU network of size $s = \text{poly}(d)$ bit precision $\frac{d}{2}$. In both cases (i) and (ii), there exists an f-PDS algorithm $\mathcal{A}$ such that for any $f : \{0,1\}^d \to \{0,1\}$ with label noise $\eta$ that satisfies either (i) or (ii) respectively, and for any $\mathcal{D}$ there exists a distribution $\mathcal{D}_f$ so that $\mathcal{A}$ f-PDS learns $f$ with error $\epsilon + \eta$ in $m = \text{poly}(d, \frac{1}{\epsilon})$ and runtime $T = \text{poly}(d, \frac{1}{\epsilon})$ with samples from $\mathcal{D}_f$.*

Note that both in the case of circuits and networks, the runime of $\mathcal{A}'$ is $\text{poly}(d, \frac{1}{\epsilon})$

**Simulating with Gradient Descent.** Now we will use the result on simulating PAC learning algorithm with non-standard gradient descent on neural networks to show that $\mathcal{A}'$ can also be turned into a gradient descent based algorithm. We will use the result Theorem 1a (for mini-batch stochastic GD) from (Abbe et al., 2021) or equivalently Theorem 17 and Remark 17 in (Abbe & Sandon, 2020) for the simulation. The theorem shows that for any dimension $d$, runtime $T$, and $q$ bits of randomness, and sample size $m$ of a PAC learning algorithm, there exists a set of neural nets $\mathcal{N}$ that depend on $d$ and the runtime $T = \text{poly}(d)$, with $p'$ parameters, a initialization with $q' = r + O(T \log b)$ bits of randomness, a poly time computable activation functions $\sigma$, and some stepsize $\gamma$, so that for training using $\rho < \min\{1/8b, 1/12\}$ approximate gradients over batches of size $b$ on the neural networks from this set $\mathcal{N}$ with stepsize $\gamma$ the following claim holds. For any $\delta > 0$ and $p' = \text{poly}(d, m, r, T, \rho^{-1}, b, \delta^{-1})$ with $T' = O(mn/\delta)$ if any such PAC learning algorithm $\mathcal{A}$ learns a distribution $\mathcal{D}$ with square loss $\epsilon + \eta$ then there is a neural net in this set that learns $\mathcal{D}$ with square loss at most $\epsilon + \eta + \delta$. Taking $\mathcal{A}$ to be the previously described PAC learning algorithm for f-PDS $\mathcal{A}'$, there exists a fixed neural network $N \in \mathcal{N}$ such that running mini batch SGD on this network we learn $f$ with respect to $\mathcal{D}_f$ with square loss $\epsilon + \eta + \delta$. Taking $\epsilon = \frac{\epsilon}{4}$ and $\delta = \frac{\epsilon}{4}$, $b = 1$, $\rho = \frac{1}{d}$, we have that for $p' = \text{poly}(d, \frac{1}{\epsilon})$, $T' = O(\frac{r^2(\log r + \log \frac{1}{\epsilon})d}{\epsilon})$, and $r' = O(\frac{r^2(\log r + \log \frac{1}{\epsilon})d}{\epsilon})$, the network $N \in \mathcal{N}$ achieves square loss at most $\epsilon/2 + \eta$ on learning $f$ over $\mathcal{D}_f$, i.e. $\mathbb{E}_{S \sim (\mathcal{D}_f, f)}[\mathbb{E}_{(\mathcal{D}_f, f)}(N(S) - f)^2] \leq \epsilon/2 + \eta$, where the expectation is over the initialization and the mini batches from $\mathcal{D}_f$.

In the theorem that we used, the training and testing distributions are the same and equal to $\mathcal{D}_f$, so we want to bound the error of the neural network $N$ on $\mathcal{D}$ based on its error on $\mathcal{D}$. Note that $\mathcal{D}_f = \frac{1}{2}\mathcal{D} + \frac{1}{2}\mathcal{D}_\theta$ so we can write

$$\mathbb{E}_{S \sim (\mathcal{D}_f, f)^m}[\mathbb{E}_{D_f}[(N(S) - f)^2)]] = \frac{1}{2}\mathbb{E}_{S \sim (\mathcal{D}_f, f)^m}[\mathbb{E}_D[(N(S) - f)^2]]$$
$$+ \frac{1}{2}\mathbb{E}_{S \sim (\mathcal{D}_f, f)^m}[\mathbb{E}_{D_\theta}[(N(S) - f)^2]].$$

Note that both of the errors are at least $\eta$, so $\mathbb{E}_{S \sim (\mathcal{D}_f, f)^m}[\mathbb{E}_{D_\theta}[(N(S) - f)^2]] \geq \eta$. Therefore, we have that $\mathbb{E}_{S \sim (\mathcal{D}_f, f)^m}[\mathbb{E}_{\mathcal{D}}[(N(S) - f)^2]] \leq 2\mathbb{E}_{S \sim \mathcal{D}_f^m}[\mathbb{E}_{D_f}[(N(S) - f)^2]] - \eta$ which implies that $N$ achieves square loss at most $\epsilon + \eta$ for leaning $f$ over $\mathcal{D}$ if we take samples from $\mathcal{D}_f$, i.e.

$$\mathbb{E}_{S \sim (\mathcal{D}_f, f)^m}[\mathbb{E}_{\mathcal{D}}[(N(S) - f)^2]] \leq \epsilon + \eta.$$

This implies that $0 - 1$ error is also bounded by $\epsilon + \eta$, which finishes the proof.

**Remark** Note that we can extend the set of allowed functions to include all $f : \{0,1\}^d \to \{0,1\}$ functions that can be encoded on $\mathrm{poly}(d)$ bits of $\{0,1\}^d$. For example, this set includes all short description functions.

$\square$

## C    LEARNING PARITY FUNCTIONS

*Proof of Theorem 4.3.* Let $S$ be the support of the parity. Recall that under random classification noise $y = \xi\chi_S(\boldsymbol{x})$ with $\xi = \mathrm{Rad}(1-\eta)$. Set the distribution $\mathcal{D}' = \frac{1}{2}\mathrm{Rad}(1-\frac{1}{d})^{\otimes d} + \frac{1}{2}\mathrm{Unif}(\{\pm 1\}^d)$ and denote $\tilde{\mathcal{D}}'$ the distribution of $(y, \boldsymbol{x})$ with $\boldsymbol{x} \sim \mathcal{D}'$. Let's compute the correlation of the label with each coordinate $x_i$, for $i = 1, \ldots, d$: using that $\mathbb{E}_{\mathrm{Rad}(1-\frac{1}{d})}[x_i] = 1 - 2/d$,

$$\mathbb{E}_{\tilde{\mathcal{D}}'}(yx_i) = (1 - 2\eta)\mathbb{E}_{\mathcal{D}'}(\chi_S(\boldsymbol{x})x_i) = \begin{cases} \frac{1}{2}(1-2\eta)(1-\frac{2}{d})^{|S|+1} & \text{if } i \notin S \\ \frac{1}{2}(1-2\eta)(1-\frac{2}{d})^{|S|-1} & \text{if } i \in S. \end{cases}$$

Then, $\Delta = \frac{1}{2}(1-2\eta)(1-\frac{2}{d})^{|S|-1}(1-(1-\frac{2}{d})^2)$ is the difference in the correlation between $i \in S$ and $i \notin S$. Note that $\Delta = \Theta((1-2\eta)/d)$.

Let $\hat{T}_i = \frac{1}{m}\sum_{j=1}^m y^j x_i^j$ be the empirical estimate of $\mathbb{E}_{\tilde{\mathcal{D}}'}[yx_i]$. We estimate the support $\hat{S}$ of the parity by taking the set of $i \in [d]$ such that $\hat{T}_i > \frac{1}{2}(\min_i \hat{T}_i + \max_i \hat{T}_i)$. By Hoeffding and union bound,

$$\Pr[\sup_{i \in [d]} |\hat{T}_i - \mathbb{E}_{\tilde{\mathcal{D}}'}(yx_i)| > t] \le 2d\exp(-2mt^2).$$

Taking $t = \frac{\Delta}{8}$, and $m = \frac{64}{\Delta^2}\log\frac{2d}{\epsilon}$, with probability $1-\epsilon$ we have that for all $i$ $|\hat{T}_i - \mathbb{E}_{\tilde{\mathcal{D}}'}(yx_i)| < \frac{\Delta}{8}$ so in particular $\hat{T}_i > \frac{1}{2}(\min_i \hat{T}_i + \max_i \hat{T}_i)$ if and only if $i \in S$. Thus we recover the parity function with number of sample $m = O(\frac{d^2\log\frac{2d}{\epsilon}}{(1-2\eta)^2})$. Therefore, for the $0-1$ loss it holds that it is at most $\epsilon + \eta$ in this case. $\square$

*Proof of Theorem 4.5.* We consider a parity function $\chi_S(\boldsymbol{x})$ on the hypercube, with $k := |S| = d/2$. Other $k$ will follow similarly. The goal is to learn this function with a two-layer neural network with activation $\mathrm{ReLu}(x) = (x)_+$. Denote $\boldsymbol{w}_S$ the vector with 1 on the support $S$ and 0 otherwise, and $g_S : \mathbb{R} \to \mathbb{R}$ a function such that $\chi_S(\boldsymbol{x}) = g_S(\langle \boldsymbol{w}_S, \boldsymbol{x}\rangle)$ (i.e., $g_S(k - 2j) = (-1)^j$). For simplicity, we assume below that $k$ is odd. Note that we can take

$$\begin{aligned} g_S(x) = & -\mathrm{ReLu}(x+k+1) + \mathrm{ReLu}(x) + 2\sum_{j=0}^{(k-1)/2}(-1)^j \cdot \mathrm{ReLu}(x+k-2j) \\ & + \mathrm{ReLu}(-x+k+1) - \mathrm{ReLu}(-x) - 2\sum_{j=0}^{(k-1)/2}(-1)^j \cdot \mathrm{ReLu}(-x+k-2j). \end{aligned} \tag{1}$$

Let $\mathcal{D}_1 = \mathrm{Rad}(1-1/d)^{\otimes d}$, i.e. each coordinate is 1 with probability $1-1/d$ and $-1$ with probability $1/d$. Let $\mathcal{D}_2$ be defined as sampling of $\boldsymbol{x}$ in the following way: Draw $k \sim \mathrm{Unif}(\{d, d-2, d-4, \ldots, -d\})$, and then sample $x$ conditioned on $\mathbf{1}^T\boldsymbol{x} = k$ uniform on the hypercube (i.e. it is a reweighted distribution on the sliced hypercube, where each slice has equal distribution). Then take

$$\mathcal{D}' = \frac{1}{2}\mathcal{D}_1 + \frac{1}{2}\mathcal{D}_2.$$

Therefore, we have that

$$\mathbb{E}_{\tilde{\mathcal{D}}'}[yx_i] = (1-2\eta)\mathbb{E}_{\mathcal{D}'}[\chi_S(\boldsymbol{x})x_i] = \frac{1-2\eta}{2}\left(\mathbb{E}_{\mathcal{D}_1}[\chi_S(\boldsymbol{x})x_i] + \mathbb{E}_{\mathcal{D}_2}[\chi_S(\boldsymbol{x})x_i]\right).$$

Similarly as above, we have

$$\mathbb{E}_{\mathcal{D}_1}[\chi_S(\boldsymbol{x})x_i] = \begin{cases} (1-2/d)^{|S|+1} & \text{if } i \notin S, \\ (1-2/d)^{|S|-1} & \text{If } i \in S. \end{cases}$$

while the expectation with respect to the second distribution yields

$$\mathbb{E}_{\mathcal{D}_1}[\chi_S(\boldsymbol{x})x_i] = \begin{cases} 0 & \text{if } |S| \text{ is even,} \\ \frac{1}{|S|-1} & \text{if } |S| \text{ odd and } i \in S \\ \frac{1}{|S|+1} & \text{if } |S| \text{ odd and } i \notin S. \end{cases}$$

We consider a two-layer neural network as follow:

$$f_{\mathsf{NN}}(\boldsymbol{x};\boldsymbol{\theta}) = \sum_{j\in[N]} a_j \mathsf{ReLu}(\langle \boldsymbol{w}_j, \boldsymbol{x}\rangle + b_j),$$

where $\boldsymbol{\theta} = (\boldsymbol{a}, \boldsymbol{W}, \boldsymbol{b}) \in \mathbb{R}^{N(d+2)}$ are the parameters of the network. We will choose as initialization $a_j^0 \sim_{i.i.d.} \mathrm{Unif}(\{\pm 1\})$, $\boldsymbol{w}_j^0 = \boldsymbol{0}$, and $b_j^0 \sim_{i.i.d.} \mathrm{Unif}(\{-d-1, -d, \ldots, d, d+1\})$. We can either use $N$ sufficiently large (but polynomial in $d$) or for simplicity we take exactly $(4d+6)$ neurons, one for each combination $(a^0, b^0)$. Note that with this choice,

$$f_{\mathsf{NN}}(\boldsymbol{x};\boldsymbol{\theta}^0) = 0.$$

We will consider a layerwise training procedure, where we first train the $\boldsymbol{w}_j$'s with one gradient step with step size $s$, followed by training the $a_j$'s with (online) SGD.

We consider a $\ell_1$-regularization on the first layer weights, with regularization parameter $\lambda$. Let's first consider what happens for population gradient: we get

$$\boldsymbol{w}_j^1 = s \cdot \rho_\lambda \left((1-2\eta)\mathbb{E}_{\mathcal{D}'}[\chi_S(\boldsymbol{x})\boldsymbol{x}\mathbb{1}[b_j \geq 0]]\right),$$

where $\rho(x;\lambda) = \mathrm{sign}(x)(|x|-\lambda)_+$ denotes soft-thresholding and applies component wise. From the computation above we can choose the regularization parameter $\lambda$ so that $\lambda$ is inbetween these values for $x_i \in S$ and $x_i \notin S$:

$$\lambda = \begin{cases} (1-2\eta)(1-2/d)^{|S|} & \text{if } |S| \text{ is even,} \\ (1-2\eta)(1-2/d)^{|S|} + \frac{1}{|S|} & \text{if } |S| \text{ odd.} \end{cases}$$

By the choice of $\lambda$ as above and stepsize $s$ as

$$s = \begin{cases} [(1-2\eta)(1-\frac{2}{d})^{|S|-1}/d]^{-1} & \text{if } |S| \text{ is even} \\ [(1-2\eta)((1-\frac{2}{d})^{|S|-1} + \frac{1}{|S|-1})/d]^{-1} & \text{if } |S| \text{ is odd.} \end{cases}$$

(note that we fix the size the parameter to be $|S| = d/2$), we get the after one step of population gradient descent, we have

$$\overline{\boldsymbol{w}}_j^1 = a_j^0 \boldsymbol{w}_S \mathbb{1}[b_j \geq 0].$$

Now, let's consider the empirical gradient update, we have by Hoeffding inequality and union bound, concentration in $\|\cdot\|_\infty$ of the gradient with $m$ samples: it holds with probability at least $1-\delta$ (where $M$ is the width of the network)

$$\sup_{j\in[M]} \|\boldsymbol{w}_j^1 - \overline{\boldsymbol{w}}_j^1\|_\infty \leq \sqrt{\frac{1}{m}\log\frac{dM}{\delta}}.$$

We will denote $\hat{\boldsymbol{W}}$ the first layer weights after this first (empirical) gradient step, and $\overline{\boldsymbol{W}}$ the first layer weights after this step with population gradient. In particular, after one population gradient step, we get the $2d+4$ neurons:

$$\mathsf{ReLu}(\langle \boldsymbol{w}_S, \boldsymbol{x}\rangle + b_j), \qquad \mathsf{ReLu}(-\langle \boldsymbol{w}_S, \boldsymbol{x}\rangle + b_j), \qquad b_j \in \{0, 1, 2\ldots, d+1\}.$$

Let's run SGD on the second layer weights $a_j$'s with ridge regularization as in (2). Let's apply Lemma C.1 to our setting. We have $p = 2d+4$. There exists a universal constant $C > 0$ such that

$$B_y = 1, \qquad B_\phi = Cd, \qquad \|\boldsymbol{a}^0\|_2^2 \leq Cd.$$

Consider $\boldsymbol{a}_{\mathsf{cert}} = (1-2\eta)\boldsymbol{a}^*$ with $\boldsymbol{a}^*$ as defined in (1): we have $f_{NN}(\boldsymbol{x}, \boldsymbol{a}_{\mathsf{cert}}, \overline{\boldsymbol{W}}) = (1-2\eta)\chi_S(\boldsymbol{x})$. It follows that

$$\begin{aligned} \|f_{\mathsf{NN}}(\boldsymbol{a}_{\mathsf{cert}}, \hat{\boldsymbol{W}}) - (1-2\eta)\chi_S\|_{L_2}^2 &= \|f_{\mathsf{NN}}(\boldsymbol{a}_{\mathsf{cert}}, \hat{\boldsymbol{W}}) - f_{\mathsf{NN}}(\boldsymbol{a}_{\mathsf{cert}}, \overline{\boldsymbol{W}})\|_{L_2}^2 \\ &\leq M^2 \|\boldsymbol{a}_{\mathsf{cert}}\|_\infty^2 \|\sigma(\langle \overline{\boldsymbol{w}}_j, \cdot\rangle + b_j) - \sigma(\langle \hat{\boldsymbol{w}}_j, \cdot\rangle + b_j)\|_\infty^2 \\ &\leq Cd^2 M^2 \sup_{j\in[M]} \|\hat{\boldsymbol{w}}_j - \overline{\boldsymbol{w}}_j\|_\infty^2. \end{aligned}$$

Thus we get

$$\mathcal{L}_\lambda(\boldsymbol{a}_{\mathsf{cert}}) \leq C\frac{d^2M^2}{m}\log\frac{dM}{\delta} + C\lambda d.$$

We choose $\lambda = \frac{(1-2\eta)^2\epsilon}{Cd}$, $s = \frac{(1-2\eta)^4\epsilon^2}{Cd^8\log(d/((1-2\eta)^2\epsilon\delta))}$ and $T = C\frac{d^9\log^2(d/((1-2\eta)^2\epsilon\delta))}{\epsilon^3}$ such that with probability at least $1 - \delta$,

$$\|(1-2\eta)\chi_S - f_{\mathsf{NN}}(\boldsymbol{a}^T; \hat{\boldsymbol{W}})\|^2_{L^2} + \lambda\|\boldsymbol{a}^T\|^2_2 \leq (1-2\eta)^2\epsilon,$$

with steps and sample complexity scaling as

$$m, T = O\left(\frac{d^9\log^2\left(\frac{d}{\epsilon\delta(1-2\eta)^2}\right)}{\epsilon^3(1-2\eta)^6}\right).$$

Let's show that this implies that

$$\|\chi_S - \mathsf{sign}(f_{\mathsf{NN}}(\boldsymbol{a}^T; \hat{\boldsymbol{W}}))\|_{L^\infty} \leq \epsilon,$$

so that it implies a test error bounded by $\epsilon + \eta$ for any test distribution. First,

$$\|f_{\mathsf{NN}}(\boldsymbol{x}, \hat{\boldsymbol{a}_T}, \hat{\boldsymbol{W}}) - f_{\mathsf{NN}}(\boldsymbol{x}, \hat{\boldsymbol{a}_T}, \overline{\boldsymbol{W}})\| \leq dM\|\hat{\boldsymbol{a}}_T\|_\infty \sup_j \|\hat{\boldsymbol{w}}_j - \overline{\boldsymbol{w}}_j\|_\infty.$$

Furthermore

$$|f_{\mathsf{NN}}(\boldsymbol{x}, \hat{\boldsymbol{a}}_T, \overline{\boldsymbol{W}}) - (1-2\eta)\chi_S(\boldsymbol{x})|$$

$$= \sum_{k=-|S|}^{k=|S|} \Pr(\langle\boldsymbol{w}_S, \boldsymbol{x}\rangle = k) \left|\sum_{j=1}^M \hat{a}_j(k + b_j)_+ - (1-2\eta)\chi_S(k)\right|^2.$$

Thus,

$$\|f_{NN}(\hat{\boldsymbol{a}}_T, \overline{\boldsymbol{W}}) - (1-2\eta)\chi_S(\boldsymbol{x})\|_\infty \leq \frac{\|f_{\mathsf{NN}}(\hat{\boldsymbol{a}}, \boldsymbol{w}^*) - (1-2\eta)\chi_S\|_{L^2}}{\sqrt{\min_{k=-|S|,\ldots,|S|} P(\langle\boldsymbol{w}_S, \boldsymbol{x}\rangle = k)}}.$$

Note that $P(\langle\boldsymbol{w}_S, \boldsymbol{x}\rangle = k) \geq \mathcal{D}_2(\langle\boldsymbol{w}_S, \boldsymbol{x}\rangle = k) = \frac{1}{d}\sum_{i=-d}^d \Pr(\langle\boldsymbol{w}_S, \boldsymbol{x}\rangle = k | \langle\boldsymbol{1}, \boldsymbol{x}\rangle = i) = \frac{1}{d}\sum_{i=0}^d \Pr(k + 1 \text{ in first } |S| \text{ coordinattes} | i \text{ total } +1 \text{ coordinates}) = \frac{1}{n}\sum_{i=k}^{d-|S|+k} \frac{\binom{|S|}{k}\binom{d-|S|}{i-k}}{\binom{d}{i}} = \frac{1}{d}\frac{d+1}{|S|+1}$. This implies that $P(\langle\boldsymbol{w}_S, \boldsymbol{x}\rangle = k) \geq \frac{1}{|S|+1} \geq \frac{1}{d+1}$. Therefore, we have that

$$\|f_{\mathsf{NN}}(\boldsymbol{x}, \hat{\boldsymbol{a}}_T, \overline{\boldsymbol{W}}) - (1-2\eta)\chi_S(\boldsymbol{x})\|_\infty \leq \sqrt{d+1}\|f_{\mathsf{NN}}(\hat{\boldsymbol{a}}, \overline{\boldsymbol{W}}) - \chi_S\|_{L^2}.$$

Note that

$$\|f_{\mathsf{NN}}(\hat{\boldsymbol{a}}, \overline{\boldsymbol{W}}) - \chi_S\|_{L^2} \leq \|f_{\mathsf{NN}}(\hat{\boldsymbol{a}}, \hat{\boldsymbol{W}}) - \chi_S\|_{L^2} + \|f_{\mathsf{NN}}(\hat{\boldsymbol{a}}, \hat{\boldsymbol{W}}) - f_{\mathsf{NN}}(\hat{\boldsymbol{a}}, \overline{\boldsymbol{W}})\|_{L^2}.$$

Combining the above bounds, we obtain

$$\|f_{\mathsf{NN}}(\hat{\boldsymbol{a}}_T, \hat{\boldsymbol{W}}) - (1-\eta)\chi_S\|_\infty$$

$$\leq \sqrt{d+1}\left(\|f_{\mathsf{NN}}(\hat{\boldsymbol{a}}, \overline{\boldsymbol{W}}) - f_{\mathsf{NN}}(\hat{\boldsymbol{a}}, \hat{\boldsymbol{W}})\|_{L^2} + \|f_{\mathsf{NN}}(\hat{\boldsymbol{a}}, \overline{\boldsymbol{W}}) - \chi_S\|_{L^2}\right)$$

$$+ dM\|\hat{\boldsymbol{a}}_T\|_\infty \sup_j \|\hat{\boldsymbol{w}}_j - \boldsymbol{w}_j^*\|_\infty$$

$$\leq \sqrt{d+1}\left(dM\|\hat{\boldsymbol{a}}_T\|_\infty \sup_j \|\hat{\boldsymbol{w}}_j - \overline{\boldsymbol{w}}_j\|_\infty + \|f_{\mathsf{NN}}(\boldsymbol{a}_{\mathsf{cert}}, \hat{\boldsymbol{W}}) - f_{NN}(\boldsymbol{a}_{\mathsf{cert}}, \overline{\boldsymbol{W}})\|_{L^2} + \frac{1}{T^c}\right)$$

$$+ \|\hat{\boldsymbol{a}}_T\|_\infty \sup_j \|\hat{\boldsymbol{w}}_j - \overline{\boldsymbol{w}}_j\|_\infty dM.$$

Therefore, we have

$$\|f_{NN}(\boldsymbol{x}, \hat{\boldsymbol{a}}_T, \hat{\boldsymbol{w}}) - \chi_S(\boldsymbol{x})\|_\infty \le (1 + 2\sqrt{d+1})\|\hat{\boldsymbol{a}}_T\|_\infty \sup_j \|\hat{\boldsymbol{w}}_j - \boldsymbol{w}_j^*\|_\infty dM + \frac{\sqrt{d+1}}{T^c}$$

$$\le (1 + 2\sqrt{d+1})\|\hat{\boldsymbol{a}}_T\|_\infty dM \sqrt{\frac{1}{m}\log\frac{M}{\delta}} + \frac{\sqrt{d+1}}{T^c}.$$

We can take $m = \text{poly}(d, \frac{1}{\epsilon}, \log\frac{1}{\delta})$ and $T = \text{poly}(d, \frac{1}{\epsilon})$ (since $M = 4d + 6$) so that with probability at least $1 - \delta$

$$\|f_{NN}(\boldsymbol{x}, \hat{\boldsymbol{a}}_T, \hat{\boldsymbol{w}}) - \chi_S(\boldsymbol{x})\|_\infty \le \epsilon.$$

This implies that the $0 - 1$ loss is bounded by $\epsilon$ with probability $\delta$, so we can take $\delta = \epsilon$ so that in expectation it's bounded by $2\epsilon$. $\qquad\square$

We recall the following useful upper bound on online SGD on a ridge regularized linear regression with features $\phi(\boldsymbol{x}) \in \mathbb{R}^p$. Denote the loss

$$\mathcal{L}(\boldsymbol{a}) = \frac{1}{2}\mathbb{E}_{\boldsymbol{x}}\left[(\mathbb{E}[y|\boldsymbol{x}] - \langle \boldsymbol{a}, \phi(\boldsymbol{x})\rangle)^2\right], \qquad \mathcal{L}_\lambda(\boldsymbol{a}) = \mathcal{L}(\boldsymbol{a}) + \frac{\lambda}{2}\|\boldsymbol{a}\|_2^2.$$

We run online SGD

$$\boldsymbol{a}^{t+1} = (1 - \lambda)\boldsymbol{a}^t + s(y^t - \langle \boldsymbol{a}^t, \phi(\boldsymbol{x}^t)\rangle)\phi(\boldsymbol{x}^t). \tag{2}$$

**Lemma C.1** (Online SGD on ridge regression (Abbe et al., 2023a))**.** *There exists a universal constant $C > 0$ such that the following holds. Suppose there exists $B_y, B_\phi \ge 1$ such that $\|\varphi(\boldsymbol{x})\|_2 \le B_\phi$ and $|y| \le B_y$, then for any $\lambda \le 1$ and $\boldsymbol{a}_{\text{cert}} \in \mathbb{R}^p$, we have*

$$\mathcal{L}(\boldsymbol{a}^t) \le \mathcal{L}_\lambda(\boldsymbol{a}_{\text{cert}}) + CB_\phi^2\left\{(1 - \lambda s)^{2t}\left(\|\boldsymbol{a}^0\|_2^2 + \frac{B_y^2}{\lambda}\right) + \log\left(\frac{t}{\delta}\right)\frac{sB_\phi^4 B_y^2}{\lambda^2}\right\}$$

*with probability at least $1 - \delta$.*

**Lemma C.2** (Positive distribution shift to learn a parity function)**.** *Consider learning a $k$-sparse parity function $f^* : \{\pm 1\}^d \to \{\pm 1\}$, $f^* = \prod_{i \in S} x_i$, $S \subset [d]$, $|S| = k$ using a two-layer neural network with a ReLU activation function*

$$f_{NN}(x; \theta) = \sum_{j \in [N]} a_j \sigma(\langle w_j, x\rangle + b_j),$$

*where $\theta = (a, W, b) \in \mathbb{R}^{N(d+2)}$. We use a layerwise training procedure, where we train $w_j$'s with one gradient step and $a_j$'s with SGD. Let $\mathcal{D}$ be the uniform distribution over $\{0,1\}^d$. There exist $N = \text{poly}(d)$, an initialization, stepsizes, a distribution $\tilde{\mathcal{D}}$ over $\{0,1\}^d$, and $m, T = O\left(\frac{d^9 \log^2(\frac{d}{\varepsilon\delta})}{\varepsilon^3}\right)$ such that after training the neural network with $T$ steps of gradient descent and $m$ samples from $\tilde{\mathcal{D}}$, we learn $f^*$ with error $\varepsilon$ with respect to $\mathcal{D}$,*

$$\mathbb{E}_{\mathcal{D}}\left((f^* - f_{NN}(x; \theta^T))^2\right) \le \varepsilon.$$

We show that noisy parities are f-PDS learnable with standard analyzable gradient descent on a neural network, namely using layerwise training procedure where we first train the bottom layer while holding the top layer fixed and then train the top layer while holding the bottom layer fixed. This assumption is standard in the theory of neural network learning literature (Abbe et al., 2023a; Barak et al., 2022; Bietti et al., 2022; Dandi et al., 2023).

**Analyzable SGD C.3** (Layerwise SGD)**.** We perform stochastic gradient descent (SGD) on a two-layer network

$$f_{\text{NN}}(\boldsymbol{x}; \boldsymbol{\theta}) = \sum_{j \in [N]} a_j \text{ReLu}(\langle \boldsymbol{w}_j, \boldsymbol{x}\rangle + b_j),$$

where $\boldsymbol{\theta} = (\boldsymbol{a}, \boldsymbol{W}, \boldsymbol{b}) \in \mathbb{R}^{N(d+2)}$ are the parameters of the network. We do layerwise training, where we first train the bottom layer while holding the top layer fixed and then train the top layer while holding the bottom layer fixed, with square loss and with stepsize $s$. We initialize hyperparameters as $a_j^0 \sim_{i.i.d.} \text{Unif}(\{\pm 1\})$, $\boldsymbol{w}_j^0 = \boldsymbol{0}$, and $b_j^0 \sim_{i.i.d.} \text{Unif}(\{-d-1, -d, \ldots, d, d+1\})$, $s = 2$, and take $N = (4d + 6)$ neurons.

**Theorem C.4** (Noisy Parities are f-PDS Learnable with Analyzable GD). *The parity class* $\mathsf{Parity}_d$ *with label noise* $\eta < \frac{1}{2}$ *is f-PDS learnable with Analyzable SGD C.3. Namely, if we train with* $\mathcal{D}' = \frac{1}{2}\mathrm{Unif}(\{\pm 1\}^d) + \frac{1}{2}\mathcal{D}_S$, *where* $\mathcal{D}_S$ *is uniform over the hypercube outside the support and on the support, all* $x_i$ *are* $+1$ *with probability* $\frac{1}{2}$ *and* $-1$ *with probability* $\frac{1}{2}$, *on* $m$ *samples from* $\mathcal{D}'$ *using* $T$ *steps of Analyzable SGD C.3 with* $m, T = O\left(\frac{d^9 \log^2(\frac{d}{\varepsilon^2(1-2\eta)^2})}{\varepsilon^3(1-2\eta)^6}\right)$, *we have that* $\mathbb{E}_{S' \sim (\mathcal{D}', f)^m}[L_{\mathcal{D}, f}(f_{NN}(x; \theta^{(T)}))] < \epsilon + \eta$.

*Proof of Theorem C.4.* We will define the distribution $\nu_S$ as the distribution where we have uniform over the hypercube outside the support, and all $x_i$ on the support equal to $+1$ with probability $1/2$, and $-1$ with probability $1/2$. We consider learning our $k$-parity using the distribution $\mathcal{D}_S$ given by

$$\mathcal{D}_S = \frac{1}{2}\left(\nu_0 + \nu_S\right).$$

We consider a two-layer neural network as follow:

$$f_{\mathsf{NN}}(\boldsymbol{x}; \boldsymbol{\theta}) = \sum_{j \in [N]} a_j \mathsf{ReLu}(\langle \boldsymbol{w}_j, \boldsymbol{x}\rangle + b_j),$$

where $\boldsymbol{\theta} = (\boldsymbol{a}, \boldsymbol{W}, \boldsymbol{b}) \in \mathbb{R}^{N(d+2)}$ are the parameters of the network. We will choose as initialization $a_j^0 \sim_{i.i.d.} \mathrm{Unif}(\{\pm 1\})$, $\boldsymbol{w}_j^0 = \boldsymbol{0}$, and $b_j^0 \sim_{i.i.d.} \mathrm{Unif}(\{-d-1, -d, \ldots, d, d+1\})$. We can either use $N$ sufficiently large (but polynomial in $d$) or for simplicity we take exactly $(4d+6)$ neurons, one for each combination $(a^0, b^0)$. Note that with this choice,

$$f_{\mathsf{NN}}(\boldsymbol{x}; \boldsymbol{\theta}^0) = 0.$$

We will consider a layerwise training procedure, where we first train the $\boldsymbol{w}_j$'s with one gradient step, followed by training the $a_j$'s with (online) SGD.

We will use concentration over the gradient at initialization to do one gradient step and noise on the population loss (we will only require $m = O(d \log(d))$ samples to do so). For simplicity, we will simply write the gist of the argument.

After one-gradient step:

$$\boldsymbol{w}_j^1 = \eta(1-2\sigma)a_j^0 \mathbb{E}_{\mathcal{D}_S}[\chi_S(\boldsymbol{x})\boldsymbol{x}] \cdot \mathbb{1}[b_j \geq 0].$$

We have

$$\mathbb{E}_{\mathcal{D}_S}[\chi_S(\boldsymbol{x})\boldsymbol{x}] = \frac{1}{2}\mathbb{E}_{\nu_S}\left[g_S(\langle \boldsymbol{w}_S, \boldsymbol{x}\rangle)\boldsymbol{x}\right] = \frac{1}{4}\left[g_S(k)\boldsymbol{w}_S - g_S(-k)\boldsymbol{w}_S\right] = \frac{1}{2}\boldsymbol{w}_S.$$

We consider (for simplicity) the step size $\eta = 2$. Hence, for $a_j^0 = +1$ and $b_j^0 \geq 0$, we get $\boldsymbol{w}_j^1 = \boldsymbol{w}_S$, and for $a_j^0 = -1$ and $b_j^0 \geq 0$, we get $\boldsymbol{w}_j^1 = -\boldsymbol{w}_S$. For the rest $\boldsymbol{w}_j^1 = \boldsymbol{0}$. Hence, after one gradient step, we get the $2d+4$ neurons:

$$\mathsf{ReLu}(\langle \boldsymbol{w}_S, \boldsymbol{x}\rangle + b_j), \qquad \mathsf{ReLu}(-\langle \boldsymbol{w}_S, \boldsymbol{x}\rangle + b_j), \qquad b_j \in \{0, 1, 2 \ldots, d+1\}.$$

The rest of the proof follows from a similar argument as the previous proof. $\qquad\square$

# D LEARNING K-JUNTAS

## D.1 D-DS-PAC LEARNABILITY BY CSQ

We recall that Correlational Statistical Queries (CSQ) algorithms (Bendavid et al. (1995); Bshouty & Feldman (2002)) access the data via queries $\phi : \mathbb{R}^d \to [-1, 1]$ and return $\mathbb{E}_{x,y}[\phi(x)y]$ up to some error tolerance $\tau$. Here, we give a construction that shows D-DS-PAC learnability of juntas by CSQ algorithms.

**Theorem D.1** (Juntas are D-DS-PAC learnable by CSQ). *Let* $\mathsf{Junta}_d^k$ *be the class of* $k$-juntas. *There exists an input distribution* $\mathcal{D}'$ *over* $\{\pm 1\}^d$ *and a CSQ algorithm such that for any* $f^* \in \mathsf{Junta}_d^k$, *and for any label noise ratio* $\eta < 1/2$, *after* $n = O(dk + 2^k)$ *queries on* $\mathcal{D}'$ *of error tolerance* $\tau \leq c\frac{1-2\eta}{2^k \log k}$, *for a universal constant* $c > 0$, *outputs an estimator* $\hat{f}$ *such that:*

$$\hat{f}(x) = f^*(x), \qquad \forall x \in \{\pm 1\}^d. \tag{3}$$

*Proof.* Let $f^* : \{\pm 1\}^d \to \{\pm 1\}$ be a $k$-junta. Fix a label noise ratio $\eta \in [0, 1/2)$. Choose $k$ distinct interpolation nodes $\mu_1, ..., \mu_k$ defined as:

$$\mu_r = \frac{1}{2}\cos((2r-1)\pi/(2k)), \qquad r = 1, ..., k. \tag{4}$$

For $r \in [k]$ let $\mathcal{D}_r := \text{Rad}(\frac{1+\mu_r}{2})^{\otimes d}$ be the product distribution on $\{\pm 1\}^d$ with $\mathbb{E}[x_i] = \mu_r$ for all $i \in [d]$. Let $\mathcal{D}' := \frac{1}{k}\sum_{r=1}^{k}\mathcal{D}_r$.

For $r \in [k]$, let $p_r(x)$ be the density of $\mathcal{D}_r$ and let $p(x) = \frac{1}{k}\sum_{r=1}^{k}p_r(x)$ be the density of $\mathcal{D}'$. Define $\phi_r(x) = \frac{1}{k}\frac{p_r(x)}{p(x)}$ and note that $|\phi_r(x)| \leq 1$, thus these are valid CSQ queries. Let $\tilde{\mathcal{D}}'$ (resp. $\tilde{\mathcal{D}}_r$) denote the joint distribution over $x \sim \mathcal{D}'$ (resp. $x \sim \mathcal{D}_r$) and $y = f(x)\xi$, with $\xi \sim \text{Rad}(1-\eta)$. For each coordinate $i \in [d]$, and for each $r \in [k]$, denote:

$$m_r := k \cdot \mathbb{E}_{\tilde{\mathcal{D}}'}[y\phi_r(x)] = \mathbb{E}_{\tilde{\mathcal{D}}_r}[y], \qquad v_{i,r} := k \cdot \mathbb{E}_{\tilde{\mathcal{D}}'}[yx_i\phi_r(x)] = \mathbb{E}_{\tilde{\mathcal{D}}_r}[yx_i]. \tag{5}$$

Define the 'denoising' function $s_i(r) := v_{i,r} - \mu_r m_r$ and note that

$$s_i(r) = (1-2\eta)(1-\mu_r^2) \cdot \sum_{T:i\in T}\hat{f}(T)\mu_r^{|T|-1} = (1-2\eta)(1-\mu_r^2)P_i(\mu_r), \tag{6}$$

where $\hat{f}(T)$ are the Fourier-Walsh coefficients of $f$ under the standard basis, and where we defined $P_i(\mu_r) := \sum_{T\subset[d]:i\in T}\hat{f}(T)\mu_r^{|T|-1}$. Note that $P_i(\mu)$ is a polynomial of degree at most $k-1$. Furthermore, $P_i(\mu) \neq 0$ if and only if $i$ is in the support of $f$. Run the following algorithm to identify the support of $f$:

- For $r \in [k]$ and $i \in [d]$ get $m_r$ and $v_{i,r}$ with $k + dk$ queries, and compute $s_i(r)$.

- For each coordinate $i$, form evaluations: $\hat{Z}_i(r)/(1-\mu_r^2)$.

- Interpolate the unique polynomial $Q_i$ of degree $< k$ that passes through $(\mu_r, \hat{Z}_i(r))$.

- Output the support estimate: $\hat{S} = \{i \in [d] : Q_i \text{ is not identically zero}\}$.

Suppose each query is perturbed by at most $\tau$. Then, $\mu_r$ and $v_{i,r}$ are each $\tau$-close to their true values, so $s_i(r)$ has error at most $2\tau$. Dividing by $1-\mu_r^2 \geq 3/4$ (since $|\mu_r| \leq 1/2$), the error in each evaluation $\hat{Z}_i(r)$ is at most $(8/3)\tau$. Note that $\mu_r, r \in [k]$, are Chebyshev interpolation nodes, thus the polynomial interpolation operator has Lebesgue constant $\Lambda_k = \theta(\log(k))$ at these nodes. Hence, the reconstructed polynomial $Q_i$ differs from $(1-2\eta)P_i$ by at most $C\tau\log(k)$, for a universal constant $C > 0$, that does not depend on $d, k$ or $f^*$. Because $f^*$ is a k-junta, its nonzero Fourier-Walsh coefficients have magnitude at least $2^{-k}$. Therefore, if $C\tau\log(k) < \frac{1}{2}(1-2\eta)2^{-k}$, then every relevant coordinate $i$ has some coefficient of $Q_i$ remaining strictly nonzero, while every irrelevant coordinate's coefficients remain below threshold. Thus, $\hat{S} = \text{supp}(f^*)$. Since $f^*$ is fully specified by its restriction to $\{\pm 1\}^{\hat{S}}$, these values can be recovered by an additional $O(2^k)$ CSQ queries $\mathbb{E}_{\tilde{\mathcal{D}}'}[\mathbb{1}(x_{\hat{S}} = z)y]$, each revealing the sign of $\mathbb{E}[y|x_{\hat{S}} = z]$ for $z \in \{\pm 1\}^{\hat{S}}$. Hence, the algorithm outputs a hypothesis $\hat{f}$ that equals $f^*$ on all inputs. $\qquad\square$

## D.2 GD ON NEURAL NETWORKS

**Theorem D.2** (Formal statement of Theorem 4.8)**.** *Let* $\mathcal{F} := \{f \in \text{Junta}_d^k : \mathbb{E}_{\mathcal{D}}[f(x)] = 0\}$, *where* $\mathcal{D}$ *denotes the uniform distribution over* $\{\pm 1\}^d$. *Consider a two-layer network with a ReLU activation function:* $f_{\text{NN}}(x; \theta) = \sum_{j\in[N]}a_j\sigma(\langle w_j, x\rangle + b_j)$, *where* $\theta = (a, W, b) \in \mathbb{R}^{N(d+2)}$. *We use a layerwise training procedure, where we train* $w_j$'s *with one gradient step and* $a_j$'s *with SGD with the covariance loss (Def D.3). For any* $\epsilon > 0$ *and any noise level* $\eta < 1/2$, *there exist* $N = O(\log(1/\epsilon)\epsilon^{-1}(1-2\eta)^{-1})$, *an initialization, stepsizes, a meta-distribution* $\mathcal{M}_{\tilde{\mathcal{D}}}$ *over distributions* $\tilde{\mathcal{D}}$ *over* $\{\pm 1\}^d$, $m = \tilde{O}(d\log(1/\epsilon)), T = O\left(\frac{1}{\epsilon^2(1-2\eta)^2}\right)$ *such that for any* $f^* \in \mathcal{F}$, *after training the neural network with* $T$ *steps of gradient descent and* $m$ *samples from* $\tilde{\mathcal{D}}$, *we have:*

$$\mathbb{E}_{\tilde{\mathcal{D}}\sim\mathcal{M}_{\tilde{\mathcal{D}}}}\mathbb{P}_{x\sim\mathcal{D}}\left(f^*(x) \neq f_{NN}(x;\theta^T)\right) \leq C(\eta + \epsilon^c),$$

*for some constants* $c, C > 0$ *that depend only on* $k$.

### D.3 Proof of Theorem D.2

**Training distribution.** For $\boldsymbol{\mu} \in [-1, 1]^d$, denote:

$$\mathcal{D}_{\boldsymbol{\mu}} := \otimes_{i \in [d]} \text{Rad}\left(\frac{\mu_i + 1}{2}\right), \tag{7}$$

where $\text{Rad}(p)$, $p \in [0, 1]$, denotes the Rademacher distribution with parameter $p$ (i.e. $z \sim \text{Rad}(p)$ if and only if $\mathbb{P}(z = 1) = 1 - \mathbb{P}(z = -1) = p$). Let us denote $\tilde{\mathcal{D}}_{\boldsymbol{\mu}} = \frac{1}{2}\mathcal{D} + \frac{1}{2}\mathcal{D}_{\boldsymbol{\mu}}$. We consider the meta-distribution $\mathcal{M}_{\tilde{\mathcal{D}}}$ such that

$$\mathcal{M}_{\tilde{\mathcal{D}}} = \text{Unif}_{\boldsymbol{\mu} \in [-1,1]^d}\left[\tilde{\mathcal{D}}_{\boldsymbol{\mu}}\right]. \tag{8}$$

In words, we first sample $\boldsymbol{\mu} \sim \text{Unif}[-1, 1]^d$, and then we draw the training samples from a mixture of the uniform (target) distribution and the shifted $\mathcal{D}_{\boldsymbol{\mu}}$.

**The covariance loss.** We use the covariance loss, defined as follows (see also Abbe et al. (2023c)).
**Definition D.3** (Covariance loss). *Let $(X, Y) = \{(x^s, y^s)\}_{s \in [m]}$ be a dataset with $x^s \in \mathcal{X}$ and $y^s \in \{\pm 1\}$, and let $\bar{y} = \frac{1}{m}\sum_{s \in [m]} y^s$. Let $\hat{f} : \mathcal{X} \to \mathbb{R}$ be an estimator. The covariance loss is defined as:*

$$L_{\text{cov}}(x, y, \bar{y}, \hat{f}) = (1 - cy\bar{y}) \cdot (1 - y\hat{f}(x))_+, \tag{9}$$

*where $c$ is a positive constant such that $c \cdot \bar{y} < 1$.*

This choice of loss is particularly convenient because it allows us to get non-zero initial population gradients on weights incident to the coordinates in the support of the target $k$-junta, and zero initial gradients outside the support, simplifying our construction. For binary classification tasks, a small covariance loss implies a small classification error (see e.g. Proposition 1 in Abbe et al. (2023c)). In particular, we note that if $\mathbb{E}_{\mathcal{D}}[f(x)] = 0$, then $L_{\text{cov}}$ corresponds to the standard hinge loss. In our case, for any training distribution $\tilde{\mathcal{D}}_{\boldsymbol{\mu}} \sim \mathcal{M}_{\tilde{\mathcal{D}}}$, we have $|\mathbb{E}_{\tilde{\mathcal{D}}_{\boldsymbol{\mu}}}[f(x)]| < 1/2$, and since we assumed $\mathbb{E}_{\mathcal{D}}[f(x)] = 0$. Therefore, we take $c = 2$. Our experiments in Figure 2 use the more common squared loss, and confirm our theoretical findings.

**Lemma D.4.** *Let $\{x^s\}_{s \in [m]}$ be i.i.d. inputs from $\mathcal{D}'$, for an input distribution $\mathcal{D}'$. Let $\bar{y} = \frac{1}{B}\sum_{s \in [B]} y^s$. If $B \geq 2C\log(d)/\zeta^2$, with probability at least $1 - d^{-C}$,*

$$|\bar{y} - \mathbb{E}_{(\mathcal{D}', f)}[y]| \leq \zeta.$$

*Proof.* By Hoeffding's inequality,

$$\mathbb{P}(|\bar{y} - \mathbb{E}_{(\mathcal{D}', f)}[y]| \geq \zeta) \leq 2\exp\left(-\frac{B\zeta^2}{2}\right) \leq 2d^{-C}.$$

$\square$

**Setup and algorithm.** Without loss of generality, we assume that $S = [k]$, where $S$ denotes the set of relevant coordinates. We assume label noise with parameter $\eta$, i.e. for each sample $s$, $y^s = f(x^s)\xi_s$, where the $\xi_s \sim \text{Rad}(1 - \eta)$ and are independent across samples. We train our network with layer-wise stochastic gradient descent (SGD), defined as follows:

$$w_{ij}^{t+1} = w_{ij}^t - \gamma_t \frac{1}{B}\sum_{s=1}^{B} \partial_{w_{ij}^t} L_{\text{cov}}(x^s, y^s, \bar{y}, f_{\text{NN}}(x^s; \theta^t)),$$

$$a_i^{t+1} = a_i^t - \xi_t \frac{1}{B}\sum_{s=1}^{B} \partial_{a_i^t} L_{\text{cov}}(x^s, y^s, \bar{y}, f_{\text{NN}}(x^s; \theta^t)),$$

where $L_{\text{cov}}$ is the covariance loss, $B \in \mathbb{N}$ is the batch size, and $\gamma_t, \xi_t \in \mathbb{R}$ are appropriate learning rates. We set $\gamma_t = \gamma\mathbb{1}(t = 0)$ and $\xi_t = \xi\mathbb{1}(t > 0)$, for $\xi, \gamma \in \mathbb{R}$, which means that we train only the first layer for one step, and then only the second layer until convergence. We initialize the first layer weights $w_{ij}^0 = 0$ for all $i \in [N], j \in [d]$, and the second layer weights $a_i^0 = \kappa$, for $i \in [N/2]$ and $a_i^0 = -\kappa$, for $i \in [N/2]$, where $\kappa > 0$ is a constant. Thus, at initialization $f_{\text{NN}}(x; \theta^0) = 0$ for all $x \in \{\pm 1\}^d$. The biases are initialized to $b_i^0 = \kappa$, for all $i \in [N]$. After the first step of training, we drawn $b_i^1 \overset{i.i.d.}{\sim} \text{Unif}[-L, L]$, where $L \geq \kappa$.

**First layer training.** Let us fix our training distribution to be $\tilde{\mathcal{D}}_{\boldsymbol{\mu}}$, and let us denote by $\tilde{\mathcal{D}}_{\boldsymbol{\mu},f}$ the joint distribution over the input $x \sim \mathcal{D}_{\boldsymbol{\mu}}$ and label $y = f(x)\xi, \xi \sim \mathrm{Rad}(1-\eta)$. Let $f(x) = \sum_{S \subseteq [d]} \hat{f}_{\boldsymbol{\mu}}(S)\chi_{S,\boldsymbol{\mu}}(x)$ be the Fourier-Walsh expansion of $f$ with orthonormal basis elements under $\mathcal{D}_{\boldsymbol{\mu}}$ (see O'Donnell (2014)), where $\chi_{S,\boldsymbol{\mu}}(x) := \prod_{i \in S} \frac{x_i - \mu_i}{\sqrt{1-\mu_i^2}}$ are the basis elements and $\hat{f}_{\boldsymbol{\mu}}(S) := \mathbb{E}_{\mathcal{D}_{\boldsymbol{\mu}}}[f(x)\chi_{S,\boldsymbol{\mu}}(x)]$ are the Fourier-Walsh coefficients of $f$. We further denote by $\hat{f}(S) := \hat{f}_{\mathbf{0}}(S)$ the coefficients under the uniform distribution. Let us first compute the initial *population gradients* of the first layers' weights, which we denote by $\bar{G}_{w_{ij}^0}$. Given our assumptions on the initialization, we have:

$$\bar{G}_{w_{ij}^0} :=$$

$$= \mathbb{E}_{\tilde{\mathcal{D}}_{\boldsymbol{\mu},f}}\left[\partial_{w_{ij}^0} L_{\mathrm{cov}}(x^s, y^s, \bar{y}, f_{\mathsf{NN}}(x^s; \theta^t))\right]$$

$$= \mathbb{E}_{\tilde{\mathcal{D}}_{\boldsymbol{\mu},f}}[a_i^0 \mathbb{1}(w_i^0 x + b_i^0 > 0)x_j \cdot y] - 2\mathbb{E}_{\tilde{\mathcal{D}}_{\boldsymbol{\mu}}}[a_i^0 \mathbb{1}(w_i^0 x + b_i^0 > 0)x_j] \cdot \bar{y}$$

$$\overset{(a)}{=} (1-2\eta) \cdot \left(\mathbb{E}_{\tilde{\mathcal{D}}_{\boldsymbol{\mu}}}[a_i^0 \mathbb{1}(w_i^0 x + b_i^0 > 0)x_j \cdot f(x)] - 2\mathbb{E}_{\tilde{\mathcal{D}}_{\boldsymbol{\mu}}}[a_i^0 \mathbb{1}(w_i^0 x + b_i^0 > 0)x_j] \cdot \mathbb{E}_{\tilde{\mathcal{D}}_{\boldsymbol{\mu}}}[f(x)]\right) + r$$

$$\overset{(b)}{=} \kappa(1-2\eta) \cdot \left(\mathbb{E}_{\tilde{\mathcal{D}}_{\boldsymbol{\mu}}}[x_j f(x)] - 2\mathbb{E}_{\tilde{\mathcal{D}}_{\boldsymbol{\mu}}}[x_j]\mathbb{E}_{\tilde{\mathcal{D}}_{\boldsymbol{\mu}}}[f(x)]\right) + r$$

$$\overset{(c)}{=} \kappa(1-2\eta) \cdot \left(\frac{1}{2}\hat{f}(\{j\}) + \frac{1}{2}\mathbb{E}_{\mathcal{D}_{\boldsymbol{\mu}}}[x_j f(x)] - \frac{1}{2}\mathbb{E}_{\mathcal{D}_{\boldsymbol{\mu}}}[x_j]\mathbb{E}_{\mathcal{D}_{\boldsymbol{\mu}}}[f(x)]\right) + r$$

$$= \frac{1}{2}\kappa(1-2\eta)(\hat{f}(\{j\}) + \cdot\mathbb{E}_{\tilde{\mathcal{D}}_{\boldsymbol{\mu}}}[f(x)(x_j - \mu_j)]) + r$$

$$\overset{(d)}{=} \frac{1}{2}\kappa(1-2\eta)(\hat{f}(\{j\}) + \hat{f}_{\boldsymbol{\mu}}(\{j\})\sqrt{1-\mu_j^2}) + r$$

$$\overset{(e)}{=:} \alpha_j + r$$

where: $(a)$ follows from Lemma D.4 with $|\eta| \leq \zeta$, $(b)$ holds because of the initialization that we have chosen, in $(c)$ and $(d)$ we used the definitions of the Fourier coefficients of $f$, and in $(e)$ we defined $\alpha_j := \frac{1}{2}\kappa(1-2\eta)(\hat{f}(\{j\}) + \hat{f}_{\boldsymbol{\mu}}(\{j\})\sqrt{1-\mu_j^2})$. The following lemma bounds the discrepancy between the effective gradients, estimated through $B$ samples, and the population gradients.

**Lemma D.5.** *Let $G_{w_{ij}^0} := \frac{1}{B}\sum_{s=1}^B \partial_{w_{ij}^0} L_{\mathrm{cov}}(x^s, y^s, \bar{y}, f_{\mathsf{NN}}(x^s; \theta^0))$ denote the effective gradient. For $\epsilon > 0$, if $B \geq 2\zeta^{-2}\kappa^2 \log\left(\frac{Nd}{\epsilon}\right)$, with probability $1 - 2\varepsilon$, then*

$$|G_{w_{ij}^0} - \bar{G}_{w_{ij}^0}| \leq \zeta,$$

*for all $i \in [N]$ and for all $j \in [d]$.*

*Proof.* We apply Hoeffding's inequality, noticing that $|G_{w_{ij}^0}| \leq 2\kappa$,

$$\mathbb{P}\left(|G_{w_{ij}^0} - \bar{G}_{w_{ij}^0}| > \zeta\right) \leq 2\exp\left(-\frac{\zeta^2 B}{2\kappa^2}\right) \leq \frac{2\varepsilon}{Nd}.$$

The result follows by a union bound. $\qquad\square$

We make use of the following Lemma, whose proof can be found in (Cornacchia et al. (2025)[Lemma 10]), which guarantees enough diversity among the hidden features after the first step.

**Lemma D.6** (Cornacchia et al. (2025), Lemma 10). *Let $\alpha_j = \frac{\kappa(1-2\eta)}{2}(\hat{f}\{j\}) + \hat{f}_{\boldsymbol{\mu}}(\{j\})\sqrt{1-\mu_j^2})$. For $\epsilon > 0$, there exists a constant $C > 0$ such that, with probability $1 - O(\epsilon^{\frac{1}{k+1}})$ over $\boldsymbol{\mu}$:*

- *For all $s, t \in \{\pm 1\}^k$, such that $s \neq t$, $\left|\sum_{j=1}^k \alpha_j(s_j - t_j)\right| \geq C\kappa(1-2\eta)\epsilon$.*

**Second layer training.** We show that the previous lemmas imply that there exists an assignment of the second layer's weights that achieves small error.

**Lemma D.7.** *Assume that $b_i \sim \text{Unif}[-L, L]$, with $L \geq \kappa$. Let $\alpha_j = \kappa(1 - 2\eta)(\hat{f}(\{j\}) + \hat{f}_{\boldsymbol{\mu}}(\{j\})\sqrt{1 - \mu_j^2})$, for $j \in [k]$. Assume $\kappa, \gamma = \theta(1)$. For $\varepsilon_1, \varepsilon_2 > 0$, if the number of hidden neurons $N > \Omega(L \log(1/\varepsilon_2)\varepsilon_1^{-1}(1 - 2\eta)^{-1})$, with probability $1 - O(\varepsilon_1^{\frac{1}{k+1}} + \epsilon_2)$, there exists a set of hidden neurons $\{i\}_{i \in [2^k]}$ and a vector $a^* \in \mathbb{R}^{2^k}$ with $\|a^*\|_\infty \leq O(\varepsilon_1^{-1}(1 - 2\eta)^{-1})$ such that for all $x \in \{\pm 1\}^d$,*

$$f(x) = \sum_{i=1}^{2^k} a_i^* \text{ReLU}\left(\gamma \sum_{j=1}^{k} \alpha_j x_j + b_i\right). \tag{10}$$

*Proof.* For all $s \in \{\pm 1\}^k$, let $v_s := \gamma \sum_{j=1}^{k} \alpha_j s_j$, and let us order the $(v_{s_l})_{l \in [2^k]}$ in increasing order, i.e. such that $v_{s_l} < v_{s_{l+1}}$ for all $l \in [2^k - 1]$. For simplicity, we denote $v_l = v_{s_l}$. By Lemma D.6, we have that with probability $1 - O(\varepsilon_1^{\frac{1}{k+1}})$ over $\boldsymbol{\mu}$, $\min_{l \in [2^k - 1]} v_{l+1} - v_l > C\gamma\kappa\varepsilon_1(1 - 2\eta)$, for some constant $C > 0$. If $N > \Omega(L \log(1/\varepsilon_2)\varepsilon_1^{-1})$, then with probability $1 - O(\varepsilon_2)$ there exists a set of $2^k$ hidden neurons $(b_l)_{l \in [2^k]}$ such that for all $l \in [2^k]$, $b_l \in (v_{l-1}, v_l)$, where for simplicity we let $v_0 = -L$. Let us define the matrix $M \in \mathbb{R}^{2^k \times 2^k}$, with entries:

$$M_{n,m} = \text{ReLU}(v_n - b_m), \qquad n, m \in [2^k].$$

Then, by construction, $M$ is lower triangular, i.e. $M_{n,m} = 0$ if $m \geq n + 1$. Furthermore, by the construction above, the diagonal entries of $M$ are non-zero. Thus, $M$ is invertible. Let us denote by $F \in \mathbb{R}^{2^k}$ the vector such that for all $l \in [2^k]$, the $l$-th entry is given by $F_l = f(s_l)$. Then, $a^* = M^{-1}F$ and

$$\|a^*\|_\infty \leq \|M^{-1}\|_\infty \|F\|_\infty$$
$$\leq C \cdot \frac{1}{\gamma\kappa\varepsilon_1(1 - 2\eta)},$$

for a constant $C > 0$. $\qquad\square$

By combining the lemmas above, we obtain that for $\kappa, \gamma, L = \theta(1)$, $B \geq \Omega(d \log(d)^2 \log(Nd/\varepsilon))$, $N \geq \Omega(\log(1/\varepsilon)\varepsilon^{-1}(1 - 2\eta)^{-1})$ with probability $1 - O(\varepsilon^c)$, for some $c > 0$, there exists a set of $2^k$ hidden neurons $\{i\}_{i \in [2^k]}$ such that

$$\forall j \in [k], i \in [2^k] : |w_{ij}^1 - \gamma\alpha_j| < \frac{1}{\sqrt{d}\log(d)};$$

$$\forall j \notin [k], i \in [2^k] : |w_{ij}^1| < \frac{1}{\sqrt{d}\log(d)};$$

and the $b_i$ are such that (10) holds. By a slight abuse of notation, let us denote by $a^* \in \mathbb{R}^N$ the $N$-dimensional vector whose entries corresponding to the hidden neurons $\{i\}_{i \in [2^k]}$ are given by Lemma D.7, and the other entries are zero. For all $i \in [N]$, let $w_i^* \in \mathbb{R}^d$ be such that $w_{ij}^* = \gamma\alpha_j \mathbb{1}(j \in [k])$. Let $\hat{\theta} = (a_i^*, w_i^1, b_i)_{i \in [N]}$ and $\theta^* = (a_i^*, w_i^*, b_i)_{i \in [N]}$. Then, for all $(x, y) \sim \tilde{\mathcal{D}}_{\boldsymbol{\mu}, f}$ we have:

$$\left|1 - y f_{\text{NN}}(x; \hat{\theta})\right| = \left|y - f_{\text{NN}}(x; \hat{\theta})\right| \tag{11}$$

$$\leq \left|y - f(x)\right| + \left|f(x) - f_{\text{NN}}(x; \theta^*)\right| + \left|f_{\text{NN}}(x; \theta^*) - f_{\text{NN}}(x; \hat{\theta})\right| \tag{12}$$

$$\overset{(a)}{\leq} 2\mathbb{1}(y \neq f(x)) + \left|\sum_{i=1}^{N} a_i^* \left(\text{ReLU}(w_i^1 x + b_i) - \text{ReLU}(w_i^* x + b_i)\right)\right|$$

where $(a)$ follows because, by Lemma D.7, the second term of (12) is zero. Note that,

$$\left| \mathrm{ReLU}(w_i^1 x + b_i) - \mathrm{ReLU}(w_i^* x + b_i) \right| \leq \left| \sum_{j=1}^{k} (w_{ij}^1 - \gamma\alpha_j)x_j \right| + \left| \sum_{j=k+1}^{d} w_{ij}^1 x_j \right| \qquad (13)$$

$$\leq \frac{2k}{\sqrt{d}\log(d)} + \left| \sum_{j=k+1}^{d} w_{ij}^1 x_j \right|. \qquad (14)$$

Now, $\sum_{j=k+1}^{d} w_{ij}^1 x_j = \sum_{j=k+1}^{d} w_{ij}^1 \mu_j + \sum_{j=k+1}^{d} w_{ij}^1 (x_j - \mu_j) := m_i + Z_i$. Then,

$$\mathbb{E}Z_i^2 = \sum_{j=k+1}^{d} w_{ij}^2 \mathrm{Var}(x_j) = O(1/\log(d)^2). \qquad (15)$$

One the other hand, since $\mu_j$ are i.i.d. in $[-1, 1]$, $m_i$ is sub-Gaussian with $\mathrm{Var}(m_i) \leq 1/\log(d)^2$. Hence,

$$\mathbb{P}(|m_i| > t) \leq 2\exp(-t^2 \log^2(d)/2). \qquad (16)$$

Setting $t = O(1/\sqrt{\log(d)})$, we get that with probability $1 - 1/d$ over $\mu$, $|m_i| < 1/\sqrt{\log(d)}$. Let $\mathcal{E}$ denote such event. Combining terms,

$$\mathbb{E}_{\mathcal{D}_\mu} \left[ \left| \sum_{j=k+1}^{d} w_{ij}^1 x_j \right| \Big| \mathcal{E} \right] = O(1/\sqrt{\log(d)}). \qquad (17)$$

Thus,

$$\mathbb{E}_{\mathcal{D}_{\mu,f}}[L_{\mathrm{cov}}(x, y, \bar{y}, f_{\mathsf{NN}}(x; \hat{\theta}))] \qquad (18)$$

$$\leq 2 \cdot \mathbb{E}_{\tilde{\mathcal{D}}_{\mu,f}} |1 - y f_{\mathsf{NN}}(x; \hat{\theta})| \qquad (19)$$

$$\leq 2 \left( 2\eta + \sum_{i=1}^{N} |a_i^*| \mathbb{E}_{\tilde{\mathcal{D}}_\mu} \left| \mathrm{ReLU}(w_i^1 x + b_i) - \mathrm{ReLU}(w_i^* x + b_i) \right| \right) \qquad (20)$$

$$\leq 2 \left( 2\eta + \|a^*\|_2 \cdot \sum_{i=1}^{N} \mathbb{E}_{\tilde{\mathcal{D}}_\mu} \left[ \left( \mathrm{ReLU}(w_i^1 x + b_i) - \mathrm{ReLU}(w_i^* x + b_i) \right)^2 \right] \right) \qquad (21)$$

$$= 4\eta + O\left( \frac{\varepsilon^{-2}(1 - 2\eta)^{-2}}{\log(d)^{1/2}} \right) \qquad (22)$$

We then use the following well-known result on the convergence of SGD on convex losses, to show that training only the second layer with a convex loss achieves small error.

**Theorem D.8** (Shalev-Shwartz & Ben-David (2014)). *Let $\mathcal{L}$ be a convex function and let $a^* \in \arg\min_{\|a\|_2 \leq \mathcal{B}} \mathcal{L}(a)$, for some $\mathcal{B} > 0$. For all $t$, let $\alpha^t$ be such that $\mathbb{E}\left[\alpha^t \mid a^t\right] = -\nabla_{a^t}\mathcal{L}(a^t)$ and assume $\|\alpha^t\|_2 \leq A$ for some $A > 0$. If $a^{(0)} = 0$ and for all $t \in [T]$ $a^{t+1} = a^t + \gamma\alpha^t$, with $\gamma = \frac{\mathcal{B}}{A\sqrt{T}}$, then*

$$\frac{1}{T} \sum_{t=1}^{T} \mathcal{L}(a^t) \leq \mathcal{L}(a^*) + \frac{\mathcal{B}A}{\sqrt{T}}.$$

We choose $\mathcal{B} = \Omega(\varepsilon^{-1}(1 - 2\eta)^{-1})$. In our case we have $\|\alpha^t\|_2 \leq 2$.

By (22), $\mathcal{L}(a^*) \leq 4\eta + \delta/2$, for $d$ large enough. Thus, to achieve loss at most $4\eta + \delta$, we need at least $T = \Omega\left(\frac{1}{\delta^2 \varepsilon^2 (1-2\eta)^2}\right)$ training steps. Since we assume $\mathbb{E}_{\mathcal{D}}[f(x)] = 0$, then $f$ cannot be a constant function, thus $c \cdot |\bar{y}| < 1 - 1/C$, for some $C > 0$. We then use Proposition 1 in Abbe et al. (2023c) to conclude that since $|\bar{y}| < (1 - 2\eta)$, then $\mathbb{1}(\mathrm{sign}(f_{\mathsf{NN}}(x; \hat{\theta})) \neq f(x)) < C(\eta + \delta)$. Since the conditions of Lemma D.7 happen with probability $1 - \epsilon^c$, for some $c > 0$, taking expectation over $\mathcal{M}_{\mathcal{D}}$ we get the theorem statement.

## E   DS-PAC AND MEMBERSHIP QUERIES: PROOFS

*Proof of Theorem 5.2.* Assume we have a D-DS-PAC learning algorithm $A$ for $\mathcal{H}$, where the algorithm draws its training data $S'$ from the auxiliary distribution $\mathcal{D}'$. By definition, the expected excess error is small: $\mathbb{E}_{S' \sim (\mathcal{D}', f)^{m(\varepsilon)}}[L_{\mathcal{D}, f}(A(S')) - L_{\mathcal{D}, f}(\mathcal{H})] \leq \varepsilon$. Then an NA-MQ algorithm can simply query those sample points $S'$ and simulate the algorithm $A$.

Moreover, learnability in the realizable setting in D-DS-PAC also implies learnability in the presence of label noise: after sampling a training set, we can resample each query sufficiently many times so that, by the coupon collector argument, every queried example is observed multiple times, and the true label can be recovered by majority vote. $\qquad \square$

*Proof of Theorem 5.3.* Let $A$ be the given NA-MQ learner. Run $A$ only to generate its query list $Q = (x_1, \ldots, x_m)$ and $m \leq m_0(\varepsilon/2)$, Let $U = \text{supp}(Q)$, so $|U| \leq m$. Define the auxiliary distribution $D'_Q$ to be the uniform distribution over $U$. Let $\mathcal{M}_{\mathcal{D}}$ be the distribution over auxiliary distributions induced by running $A$ (which can be randomized) to generate its query list $Q$ and setting $D'_Q$ uniform over $Q$.

Choose failure probability $\delta = \varepsilon/2$. By the coupon-collector bound, if $\tilde{m} \geq \Omega(|U|(\log|U| + \log(1/\delta)))$, then with probability at least $1 - \delta$ every $u \in U$ appears in $S$. On this event, the learner reconstructs the full labeled set $\{(u, f(u)) : u \in U\}$ (taking the first occurrence for each $u$). It then runs $A$ on this set to obtain $\hat{h}$.

Conditioned on full coverage of $U$, the distribution of the reconstructed labeled query list is exactly the NA-MQ transcript that $A$ expects. Thus

$$\mathbb{E}\left[L_{\mathcal{D}, f}(\hat{h}) \middle| \text{coverage}\right] \leq \inf_{h \in \mathcal{H}} L_{\mathcal{D}, f}(h) + \varepsilon/2.$$

On the failure event (probability $\leq \delta$), the learner outputs a fixed $h_0 \in \mathcal{H}$, which adds at most $\delta$ to the error. Therefore,

$$\mathbb{E}[L_{\mathcal{D}, f}(\hat{h})] \leq \inf_{h \in \mathcal{H}} L_{\mathcal{D}, f}(h) + \varepsilon/2 + \delta \leq \inf_{h \in \mathcal{H}} L_{\mathcal{D}, f}(h) + \varepsilon.$$

Using $|U| \leq m_0(\varepsilon/2)$ and $\delta = \varepsilon/2$, we have

$$\tilde{m} = O\Big(m_0(\varepsilon/2)\big(\log m_0(\varepsilon/2) + \log(1/\varepsilon)\big)\Big).$$

This gives the RDSPAC sample bound. The procedure is polynomial-time assuming $A$ is.

**Deterministic case.**   If $A$ is deterministic, then $Q$ (and hence $U$) is fixed given $\mathcal{D}$, so the auxiliary distribution can be chosen deterministically as $D' := D'_Q$. This yields DDSPAC learnability with the same sample bound.

$\qquad \square$

## F   CLASSIFYING HYPOTHESIS CLASSES ACROSS FRAMEWORKS

In this section, we elaborate on the classification of hypothesis classes within the PDS framework and related learning models.

- **Parities**: For $a \in \{0, 1\}^d$, define the parity $\chi_a(x) = \bigoplus_{i=1}^d a_i x_i = \langle a, x \rangle \mod 2$. The parity class over $d$ bits is defined as $\text{Parity}_d = \{\chi_a : a \in \{0, 1\}^d\}$. We consider also $k$-sparse parities, $\text{Parity}_d^k = \{\chi_a : a \in \{0, 1\}^d, \|a\|_0 = k\}$.

- **Juntas**: For $k \leq d$, the $k$-junta class is $\text{Junta}_d^k = \{f : \{0, 1\}^d \to \{0, 1\} : \exists S \subseteq [d], |S| \leq k, \exists g : \{0, 1\}^S \to \{0, 1\} \text{ s.t. } f(x) = g(x_S)\}$.

- Boolean circuits: Let $B$ be the set of gates $\{\text{AND, OR, NOT}\}$. A Boolean circuit $C$ on $d$ inputs is a finite directed acyclic graph (DAG) with input nodes $x_1, \ldots, x_d$, internal nodes labeled by gates in $B$, and one output node. For $x \in \{0, 1\}^d$, values propagate along edges

to define $f_C(x) \in \{0, 1\}$. The size of the circuit is the number of gate nodes. Let $\mathsf{Circuit}_d^{t,s}$ be the set of all Boolean functions on $d$ variables that are computable by Boolean circuits of depth at most $t$ and size at most $s$.

- **Disjunctive normal form (DNF)**: A term is a conjunction of literals. A DNF with $m$ terms is a function $f(x) = T_1(x) \vee \cdots \vee T_m(x)$ where each $T_j$ is a term over $\{x_1, \ldots, x_d\}$. The class of DNF formulas with at most $s(d)$ terms over $d$ variables is $\mathsf{DNF}_d^s = \{f : \{0, 1\}^d \to \{0, 1\} : \exists m \leq s(d) \text{ s.t. } f(x) = \bigvee_{j=1}^m T_j(x)\}$.

- **Decision Tree**: A (binary) decision tree on variables $x_1, \ldots, x_d$ is a rooted tree whose internal nodes are labeled by variables and whose edges correspond to outcomes $x_i = 0$ or $x_i = 1$, where leaves are labeled by outputs in $\{0, 1\}$. It computes the function given by evaluating the tested variables along the unique root-to-leaf path. Let $\mathsf{DT}_d^{t,s}$ be the set of all Boolean functions on $d$ variables that are computed by (binary) decision trees of depth at most $t$ and size at most $s$.

- **Sparse functions**: Let $f : \{0, 1\}^d \to \{-1, 1\}$ be a Boolean function with Fourier expansion $f(x) = \sum_{S \subseteq [d]} \hat{f}(S) \chi_S(x)$, where $\chi_S(x) = (-1)^{\sum_{i \in S} x_i}$ are the parity functions. We say that $f$ is $s$-*sparse* if the number of nonzero Fourier coefficients is at most $s$, i.e., $\left|\{S \subseteq [d] : \hat{f}(S) \neq 0\}\right| \leq s$.

- **Deterministic finite automaton (DFA)**: Let $\Sigma = \{0, 1\}$. A DFA is a tuple $M = (Q, \Sigma, \delta, q_0, F)$ with finite state set $Q$, transition function $\delta : Q \times \Sigma \to Q$, start state $q_0 \in Q$, and accept set $F \subseteq Q$. For length $d$, $M$ induces a Boolean function $f_M : \{0, 1\}^d \to \{0, 1\}$ by $f_M(x_1, \ldots, x_d) = \mathbb{I}\{\delta^{(d)}(q_0, x_1, \ldots, x_d) \in F\}$, where $\delta^{(d)}$ extends $\delta$ to strings. Let $\mathsf{DFA}_d^s$ be the set of all Boolean functions on $d$ variables that are computed by a DFA with at most $s$ states.

**Juntas.** $\log(d)$-Juntas are known to be deterministically NA-MQ learnable in the realizable setting (Bshouty & Costa, 2016). Because deterministic NA-MQ learners fix their queries in advance, random classification noise can be overcome by repeating each query sufficiently many times (ensuring, by the coupon collector argument, that every queried example is observed multiple times) and taking the majority label. This in turn implies that noisy juntas are also D-DS-PAC learnable. We provide a direct proof of this fact by a different method (using correlation statistical queries), presenting a simple Correlational Statistical Query (CSQ) algorithm for D-DS-PAC learning juntas with random label noise.

In the agnostic setting, this class is learnable in the random walk model Arpe & Mossel (2008), and hence also in NA-MQ. Since we established the equivalence between R-DS-PAC and NA-MQ, it follows that juntas are R-DS-PAC learnable in the agnostic setting as well.

**DNFs.** Feldman (2007) showed that DNFs are learnable under the uniform distribution in the realizable case using non-adaptive membership queries. Consequently, DNFs are also R-DS-PAC learnable, including in the presence of random classification noise. It remains an open question whether this class is D-DS-PAC learnable. Bshouty et al. (2005) further established that DNFs are learnable in the random walk model, which is a more "passive" setting than NA-MQ, while Bartlett et al. (1994) had earlier shown the result for 2-term DNFs. Prior to these works, DNFs were known to be A-MQ learnable under the uniform distribution (Blum et al., 1994; Jackson, 1997). Most recently, Alman et al. (2025) gave a quasi-polynomial time algorithm for DNFs using (adaptive) membership queries that applies to arbitrary distributions.

**Decision trees.** Bshouty (2018) showed that decision trees of logarithmic depth in the dimension are distribution-free learnable in the realizable case using non-adaptive membership queries. Consequently, they are also R-DS-PAC learnable, including in the presence of random classification noise. It remains an open question whether this class is D-DS-PAC learnable. Bshouty et al. (2005) further established that decision trees are learnable in the random walk model. This class has long been known to be learnable using adaptive membership queries (Kushilevitz & Mansour, 1993), with subsequent improvements by Bshouty & Haddad-Zaknoon (2019). In the agnostic setting, decision trees are learnable under the uniform distribution in A-MQ (Gopalan et al., 2008).

**Sparse functions.** Kushilevitz & Mansour (1993) showed that for all functions $f : \{0, 1\}^n \to \{0, 1\}$ that are $\epsilon$ approximated by a $t$ sparse function in $L_2$, there exists a randomized polynomial

time algorithm using (adaptive) membership queries that on input $f$ and $\delta$ returns $h$ such that with probability $1 - \delta$, $h$ $O(\epsilon)$ approximates $f$ in $L_2$. This implies that sparse functions are A-MQ learnable for the uniform distribution.

**DFAs.** A seminal result by Angluin (1987) showed that this class can be learned using adaptive membership queries. Whether it is also NA-MQ learnable, or whether a separation exists, remains an open question (to the best of our knowledge).

**Circuits.** Learning general circuits remains computationally intractable even with membership queries. For instance, it is known that constant-depth Boolean circuits ($AC^0$) and threshold circuits ($TC^0$) are hard to learn under the uniform distribution, even in the A-MQ model (Kharitonov, 1993). While the precise hardness assumptions for Boolean circuits are not always considered "standard", threshold circuits already provide strong evidence of intractability: depth-4 TC circuits can implement pseudorandom functions (Krause & Lucks, 2001; Naor & Reingold, 2004), which implies hardness of learning with membership queries for every non-trivial input distribution. Moreover, recent work (Chen et al., 2022, Section 6) shows that such pseudorandom function constructions also yield hardness results for learning real-valued neural networks of depth 5 or 6 under natural distributions such as the Gaussian. Together, these results highlight that circuit classes capable of expressing pseudorandom functions are not learnable, even with powerful query access, unless one is willing to break widely believed cryptographic assumptions.

# G EXPERIMENT DETAILS AND ADDITIONAL EXPERIMENTS

## G.1 EXPERIMENT DETAILS

All experiments were performed using the PyTorch framework (Paszke et al. (2019)) and they were executed on NVIDIA Volta V100 GPUs. Each experiment is repeated 5 times and we plot the mean; shaded regions denote $\pm 1$ standard deviation.

**Architecture.** For the results presented in the main, we used a 2-layer MLP architecture trained by SGD with the square loss. In this Section, we also present some experiments obtained with a 4-layer MLP trained by SGD with the squared loss.

- **2-layer MLP.** This is a fully-connected architecture, with 1 hidden layer of 1024 neurons, and ReLU activation.
- **4-layer MLP.** This is a fully-connected architecture of 3 hidden layers of neurons of size $512, 512, 64$, and ReLU activation.

We use PyTorch's default initialization, which initializes weights of each layer with $\mathrm{Unif}[\frac{1}{\sqrt{\dim_{\mathrm{in}}}}, -\frac{1}{\sqrt{\dim_{\mathrm{in}}}}]$, where $\dim_{\mathrm{in}}$ is the input dimension of the corresponding layer.

**Training procedure.** We consider the $\ell_2$ loss: $L_{\ell_2}(\hat{y}, y) := (\hat{y} - y)^2$. We sample fresh batches of samples at each iterations. We stop training either when the training loss is less than $0.01$, or when $10^6$ iterations are performed. We compare PDS with no-PDS, where for PDS is defined as follows for parities and juntas:

- **PDS for Parities**: We select samples from $\mathcal{D}' = \frac{1}{2}\mathrm{Unif}\{\pm 1\}^d + \frac{1}{2}\mathrm{Rad}(1 - 1/d)^{\otimes d}$. The test error is computed on the uniform distribution.
- **PDS for Juntas**: For each experiment, we independently draw $\boldsymbol{\mu} \sim \mathrm{Unif}[-1, 1]^{\otimes d}$. We select training samples from $\mathcal{D}' = \frac{1}{2}\mathrm{Unif}\{\pm 1\}^d + \frac{1}{2} \otimes_{i \in [d]} \mathrm{Rad}((\mu_i + 1)/2)$. The test error is computed on the uniform distribution. The test error is computed on the uniform distribution.

In the **no-PDS** experiments, we select both training and test samples from $\mathrm{Unif}\{\pm 1\}^d$, for both parities and juntas. In all experiments, the test-set is of size 8192.

**Hyperparameter tuning.** The primary goal of our experiments is to conduct a fair comparison between PDS and no-PDS training. Thus, we did not engage in extensive hyperparameter tuning. We tried different batch sizes and learning rates, and we did not observe significant qualitative difference. We chose to report the experiments obtained for a standard batch size of 64 and a learning rate of $0.01$ for 2-layer MLP and of $0.05$ for 4-layer MLP.

## G.2 Additional Experiments

We complement the main experiments with two additional figures. Figure 3 uses the same PDS training distribution as Figure 1 but on a sparse (rather than dense) parity, and again shows clear gains from PDS. Figure 4 studies juntas under the same PDS as Figure 2, now with a 4-layer network. Interestingly, in the left plot and for no-PDS, some seeds learn faster with label noise than without; nonetheless, across seeds and noise levels, PDS consistently yields faster and more reliable learning even with the deeper architecture.

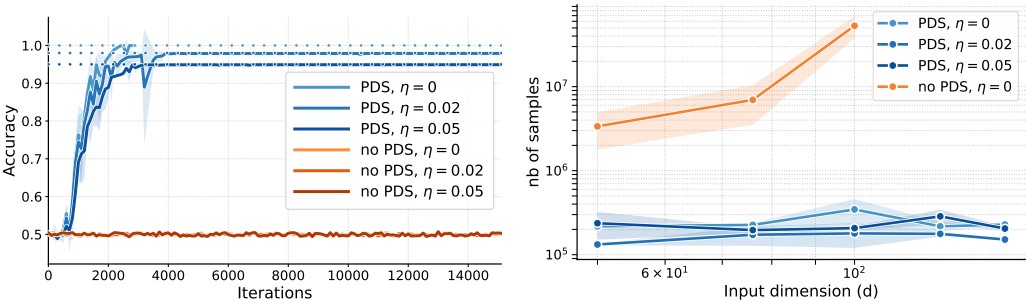

Figure 3: **Sparse parity with noise.** We compare PDS and standard (no-PDS) learning for a 5-parity with label noise $\eta \in \{0, 0.02, 0.05\}$. (Left) For $d = 50$, we plot the test accuracy on $\mathcal{D}$ versus gradient descent steps for a 4-layer ReLU network trained with SGD (batch size $b = 64$) on fresh samples from $\mathcal{D}'$ (PDS) and from $\mathcal{D}$ (no PDS). Dotted lines show Bayes accuracy (i.e., $1 - \eta$). (Right) We plot the sample complexity to reach within 0.01 of Bayes error versus input dimension. We report the simulations that converged within $10^6$ training steps. In both figures, we see that PDS training is markedly more efficient.

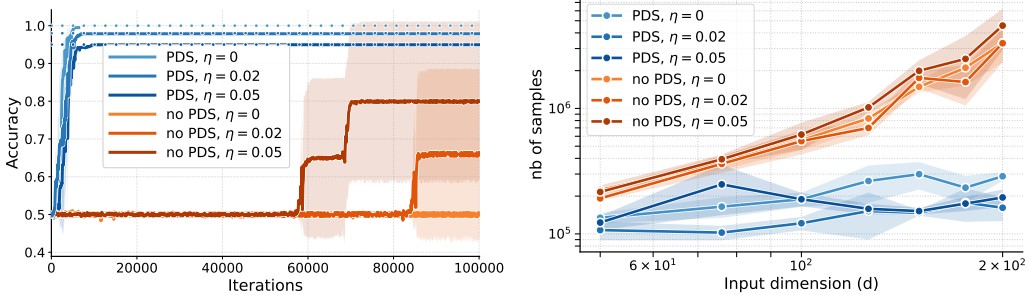

Figure 4: **Sparse juntas with noise.** We compare PDS and standard (no-PDS) learning for $f_9$ (see (4)) with label noise $\eta \in \{0, 0.02, 0.05\}$. (Left) For $d = 50$, we plot the test accuracy on $\mathcal{D}$ versus gradient descent steps for a 4-layer ReLU network trained with SGD (batch size $b = 64$) on fresh samples from $\mathcal{D}'$ (PDS) and from $\mathcal{D}$ (no PDS). Dotted lines show Bayes accuracy (i.e., $1 - \eta$). (Right) For $f_7$, we plot the sample complexity to reach within 0.01 of Bayes accuracy versus input dimension. In both figures, we see that PDS training is markedly more efficient.

