# OpenReview forum: "Positive Distribution Shift as a Framework for Understanding Tractable Learning"
_ICLR.cc/2026/Conference — Submitted to ICLR 2026_

### Official Review · Reviewer_Acce · 2025-10-26

**Soundness:** 3
**Presentation:** 3
**Contribution:** 3
**Rating:** 6
**Confidence:** 3

**Summary:**

Typically, training on a different distribution $D'$ than the target distribution $D$ (a covariate shift) is seen as problematic, but the authors in this paper introduce Positive Distribution Shift (PDS): the idea that with a well-chosen $D'$, learning a target function on $D$ can actually become easier.
In other words, the paper argues that many computationally hard learning problems can be made tractable simply by picking a clever training distribution, while keeping the model and algorithm (e.g., standard SGD on a neural network) unchanged.

The authors provide a theoretical framework for PDS and demonstrate its power through examples. They formalize several variants of PDS learning, including cases where the training distribution may depend on the target function or not.
Using these definitions, they show that certain hypothesis classes which are computationally infeasible to learn under the same train-test distribution become efficiently learnable under an appropriate distribution shift.
These theoretical findings are complemented by empirical evidence: the authors train standard neural networks on strategically biased distributions and observe successful generalization to the original target distribution, confirming that PDS can markedly improve learning efficiency in practice.

Finally, the paper draws a connection between PDS and membership query learning.
It shows that designing a training distribution is analogous to a non-adaptive way of querying an oracle for labels, thereby unifying PDS with classical active learning concepts.

**Strengths:**

- **Originality:** The paper introduces a fresh viewpoint on distribution shift. Instead of treating distribution shift as a challenge to overcome, it is framed as a resource to exploit for easier learning. This positive framing of distribution shift is, to the best of my knowledge, original and insightful. It connects to the intuition behind practices like pre-training or curriculum learning, but provides a formal lens to understand them.
- **Theoretical contributions:** The paper is thorough in developing a theoretical foundation for PDS. It defines multiple variants of the PDS learning framework (function-dependent PDS, as well as deterministic and randomized distribution-shift PAC learning) with clear formalisms. Building on these definitions, the authors prove several non-trivial results. Notably, they show that classes previously deemed intractable under standard PAC learning become tractable with an appropriate choice of training distribution.
- **Clarity:** Overall, the paper is written clearly and is well-structured. The authors do a commendable job explaining the motivation and implications of PDS. In terms of writing, the technical content is dense but generally understandable, and the authors have provided intuition alongside formal statements (for example, explaining in words how a biased distribution reveals parity bits via correlations).
- **Significance:** The framework and results presented could have significant implications for learning theory and practice. By showing that “all functions are easy, with the right training distribution” (as posed in Section 3) in certain formal senses, the paper provides a possible explanation for the effectiveness of strategies like active learning, curriculum design, and synthetic data augmentation.

**Weaknesses:**

- **Reliance on strong or unrealistic assumptions in some settings:** Some of the theoretical formulations, particularly the function-dependent PDS (f-PDS) scenario, assume knowledge that would not be available in practice. In the f-PDS framework, the training distribution $D'$ is allowed to depend on the target function $f$ itself. This is a very powerful setting. Indeed, under f-PDS the authors note one can even learn all poly-size circuits with label noise (by encoding $f$ into $D’$), but it borders on a cheat, since if one knows enough about $f$ to craft $D'$, the learning problem becomes trivial. While the paper’s main focus soon shifts to more realistic variants (where $D'$ may depend on the target distribution $D$ but not on the unknown function $f$), the strongest positive results often stem from assumptions like f-PDS or the learner having partial knowledge (e.g. knowing the parity’s sparsity). This limits the direct applicability of those results.
- **Specialized training procedures for provable results:** In the cases where the paper tackles hard classes without assuming knowledge of $f$ (notably in the DS-PAC setting), the proposed learning methods sometimes rely on non-standard or tailored techniques. For example, the gradient-based algorithms used in the proofs are “stylized” in that they require $\ell_1$ regularization or carefully chosen hyperparameters that depend on the target function’s complexity (such as the sparsity $k$ of a parity). Theorem 4.5, for instance, shows PDS learnability of parity by gradient descent but only when the network is trained with a special initialization and regularization tuned to the parity’s $k$. This undermines, to a degree, the claim that standard SGD on a standard architecture suffices.
- **Limited scope of empirical evaluation:** The experiments provided, while valuable, are restricted to synthetic boolean function tasks (parities and juntas). These tasks are canonical in theoretical computer science, but they are far from realistic machine learning applications. It remains an open question how well the PDS framework would translate to, say, image classification, natural language, or other complex domains. The paper would be stronger if it at least discussed or hypothesized about this translation. For example, can we view certain data augmentation strategies or curriculum learning in deep learning as creating a “positive distribution shift”? The authors hint at this connection in motivation but do not demonstrate it. I encourage the authors to add at least a discussion section contemplating how one might *operationalize* PDS in practical settings (e.g. how to simulate the “well-chosen $D'$” when you have limited access to $D$). This would help readers appreciate the potential impact beyond curated boolean problems.

**Questions:**

- **Practical identification of $D’$:** In a real-world scenario, how would one go about finding a “good” training distribution $D'$ without knowing the target function in advance? The theoretical results often assume either an omniscient choice of $D'$ (in f-PDS, depending on $f$) or at least knowledge of the class’s structure (e.g. knowing that a parity has sparsity $k$ or a junta has $k$ relevant variables). In practice, one may only have a small sample from the target distribution $D$ or some domain knowledge. Could the authors discuss strategies or heuristics to approximate a positive distribution shift in absence of full knowledge? For example, might one use a search procedure or adaptive data collection to converge on a helpful $D’$.
- **Generality to complex domains:** How do the authors envision applying the PDS framework to more complex or structured prediction problems (images, text, etc.)? The current examples are boolean functions with fairly clear-cut “structure” that $D'$ can exploit (e.g. biases on specific bits). In more complex tasks, a positive distribution shift might correspond to something like focusing on easier sub-tasks or higher signal-to-noise data. Are there any preliminary experiments or observations on tasks beyond parity/juntas that the authors could share to illustrate PDS in action?

---

> ### Author Response · Authors · 2025-11-21
>
> We thank the reviewer for their positive feedback and knowledgeable comments and suggestions.
>
> **Regarding operationalizing the framework and practical identification of D’**: In lines 429–442 (Summary Section) we highlight what insight we gain and impact we see (we see now that this is phrased focusing on f-PDS, but we should have written this to more explicitly refer to both frameworks, and will update this in a revision):
>
> -- The first is purely theoretical, though we still think it valuable and important, and concerns a change in how we think about computational hardness. As we state it (lines 438-441), we call for a change from asking “which functions are tractably learnable” to asking “for any function, under what training distributions is it learnable”.  This is operational in terms of what research we do, and has implications on research, though not directly on training.
>
> --The second is that **understanding which training distributions are good and allow learning (in particular using SGD), can help us in choosing data sources for training**, or understand when learning might succeed.  The way we view the realistic training procedure, which is in line with current practices, is not that the learning rule can mechanistically choose an arbitrary training distribution D’ (in the same way that a learner can choose query points in MQ or active learning).  Rather, part of the training procedure (as it actually happens in real life) is that we seek training data (which we can think of as coming from a training distribution) which is available to us, or that can be collected, and that is “good”.  This is currently done ad-hoc in a non principled way, and there has also been recent interest in empirical procedures for such training-set-selections (e.g. choosing the best training distributions in Pile, https://arxiv.org/abs/2403.16952, https://arxiv.org/abs/2407.01492, https://arxiv.org/abs/2305.10429, https://arxiv.org/abs/2507.09404).  Another example, in an even more complex domain, is https://arxiv.org/abs/2405.14838 where the training distribution involves a chain-of-thought, but this is “internalized” and prediction is made without using a chain-of-thought. We view our results, and more broadly the study of PDS which we are hoping to initiate here, as **giving principled guidance to “data collectors” in terms of what properties of a distribution (i.e. potential training data) would make it a good training distribution**.  We hope our work will give insight into what makes a good training distribution.  The main insight so far is that we should look for training distributions that more directly highlight correlations of intermediate computations and the target (individual features of the parity, or more broadly perhaps intermediate features).  This insight can be applied, e.g., to the internalized chain-of-thought example mentioned earlier, understanding it through a shifted distribution that provides more direct correlation to internal units of the final learned network.
>
> Response to other questions and comments:
>
> **Regarding strong and unrealistic assumptions, especially for f-PDS**:  We completely agree with you and say explicitly that Theorem 3.2 amounts to “cheating” (in lines 182 and again in line 430), and is used to build towards Question 3.3 and the DS-PAC framework
>
> **Specialized training procedures for provable results**: We base our conclusions about SGD on a combination of empirical evidence (e.g. Figures 1 and 2) and rigorous analysis of stylized/simplified models.  We wish we could give a satisfying theoretical analysis of SGD as it is actually used, but this seems well beyond reach of current theoretical tools, and essentially all rigorous analysis of SGD on deep networks use simplifications similar to ours e.g. https://arxiv.org/pdf/2302.11055, https://arxiv.org/pdf/2210.15651, https://arxiv.org/abs/2207.08799, https://arxiv.org/abs/2305.18270. We are intentionally being very explicit about this (e.g. in lines 85-90).
>
> **Regarding knowing the sparsity level of the parity**: this is a simplification used in the theoretical results, but is not needed empirically.  From a theoretical (and practical) perspective, even with a known sparsity level learning a parity is still hard, and so we could have just discussed parities of fixed sparsity, e.g. of exactly half the coordinates.
>
> **Generality to complex domain**: While our concrete examples focus on canonical Boolean hard classes, the PDS framework itself makes no assumption on the input space and is intended to apply equally to images, text, and other structured data. For concreteness, and to be able to develop ideas rigorously and build a solid understanding, we also focused on supervised learning and shifting the input distribution, but we view the idea of PDS as applying much more broadly to other types of tasks and training data.  We will add a discussion of these points in the revised version of the paper.

---

### Official Review · Reviewer_93oB · 2025-10-29

**Soundness:** 1
**Presentation:** 1
**Contribution:** 1
**Rating:** 2
**Confidence:** 3

**Summary:**

The paper appears to argue that carefully selected out-of-distribution (OOD) training data can actually improve in-distribution performance, introducing the concept of “positive distribution shift.” However, the authors state that the benefit of such shifts is “often computational rather than statistical,” which is not clearly explained. It remains unclear what specific computational advantages they refer to or how these differ from statistical improvements.

**Strengths:**

No strength found.

**Weaknesses:**

1. Clarity of presentation: It is unclear whether the section labeled “Introduction” serves as a motivation, problem formulation, or a mix of both. The narrative does not clearly define the research question or the contribution.

2. Ambiguity in notation and exposition: The description in Lines 57–74 (Page 2) is difficult to follow. The notation appears unorthodox and lacks explanation of key terms and symbols. This section is crucial for understanding the rest of the paper, yet it remains opaque.

Given that these issues make the core idea and mathematical formulation unclear, I was unable to evaluate the remainder of the paper in a meaningful way.

**Questions:**

I would carefully rewrite notations and claims before sending it out for further peer review.

---

> ### Author Response · Authors · 2025-11-18
>
> Lines 57-74 use standard notation and terminology and other reviewers had no problem understanding them. The review does not specify what term or notation the reviewer had difficulty with. The terms we rely on are: conditional distribution, random variable, joint distribution, learning rule, training set, hypothesis class, and runtime. Additional terms explicitly defined are: predictor, realizable, random classification noise, and error. The notation follows standard mathematical notation for probabilities, sets, and functions and is very similar to standard textbooks, e.g. “Understanding Machine Learning: From Theory to Algorithms” by Shai Ben-David and Shai Shalev-Shwartz.
>
> The research question is summarized in Lines 79-90 (relying on the definition and motivation above it, which is also summarized in the abstract), and the contributions are recapped in Lines 429-449. Again, other reviewers had no problem understanding this.

---

> ### Comment · Reviewer_93oB · 2025-11-19
> **Final Reviewer Response**
>
> Thank you for the clarification. I would like to respond to the points you raised, since several of your statements do not address the substance of my review and in some cases are factually inaccurate.
>
> 1. My concern was not only about the meaning of conditional distribution notation, but also about the ambiguity and presentation in Lines 57–74. You write that the notation uses “standard conditional distribution, random variable, joint distribution,” etc. I am familiar with these concepts. The issue is the way you defined
>   $$y | x = f(x)$$
> in the main text.
>
> This line mixes:
>
>  - y as a random variable,
>  - x as a random variable and
>  - f(x) as a deterministic function with no explicit declaration of the underlying probability space.
>  - Random variables are usually uppercase letters.
>
> The notation y | x = f(x) is ambiguous. Again f(x) is defined as a target function in the abstract! But you use f(x) to denote both:
>
> (a) a function representing a deterministic label, and
> (b) a conditional distribution law (your footnote explains this, but only after the fact).
>
> This creates a type mismatch:
>
>  - If y is a random variable, then  I do not know what y|x stands for. Is it another random variable? But you mention this as distribution.
>
>  - If f(x) is deterministic, then y = f(x) is not a conditional distribution at all.
>
> Your text uses the same object to denote two semantically incompatible roles. This is exactly the ambiguity I pointed out. Stating that it follows “standard notation” does not resolve the mismatch, if anything, it reinforces that the notation as written diverges from standard PAC presentations (e.g., Shalev-Shwartz & Ben-David).
>
> 2. You did not address the specific lines I referenced. I explicitly referenced: Lines 57–74 (Page 2) where the confusion originates. Your response instead gives a general statement about “standard notation” without addressing:
>
>  - the inconsistent use of lowercase random variables,
>
>  - the lack of formal definition for the joint distribution (D,f),
>
>  - the implicit assumption that f induces both a deterministic and stochastic law.
>
> If you believe the notation is correct as written, you should be able to point to a precise location where the semantics of y, x, and f as random variables / deterministic labels / conditional laws are fully defined. You did not do that.
>
> 3. Mentioning other reviewers’ understanding is not relevant. In your response you state:
> “Other reviewers had no problem understanding them.”
> This is not a scientific rebuttal. Review validity is not determined by majority vote. If a section of mathematical exposition is ambiguous, it is ambiguous regardless of how many readers interpreted it differently. Please focus on rectifying the ambiguity itself rather than appealing to other reviews.
>
> 4. You did not address the structural clarity issue. My review explicitly stated: The “Introduction” mixes motivation, problem formulation, and contributions without clear boundaries. The flow of notation and definitions appears before the reader has a conceptual framing of the problem. Several definitions (e.g., “training distribution,” “target distribution,” and “PDS variants”) appear abruptly without guiding narrative or roadmap. There is no relevant literature review at all! Your response did not address any of these issues.
> Again, saying “the research question is in lines 79–90” does not respond to the concern that the structure of the introduction is difficult to follow.
>
> 5. Dense mathematical exposition in the introduction is not acknowledged. Your response incorrectly implies that my concern was about lack of math literacy. What I wrote was that the introduction presents dense mathematical definitions before establishing high-level clarity, creating unnecessary cognitive load. This is a writing issue, not a technical one. You did not respond to this point at all.

---

### Official Review · Reviewer_9Uzc · 2025-11-01

**Soundness:** 2
**Presentation:** 3
**Contribution:** 3
**Rating:** 8
**Confidence:** 2

**Summary:**

The authors propose a novel notion of positive distribution shift (PDS) learnability and investigate its properties. PDS learnability refers to the learnability when the learner observes samples from a well-behaved distribution to construct a classifier for the test distribution. The authors first analyze PDS learnability in situations where the training distribution can be chosen depending on the underlying labeling rule, and show that any polynomial-size circuit can be PDS learnable by a feed-forward neural network with polynomial runtime. Under the non-PDS setup, even constant-depth circuits are not learnable. Subsequently, the authors elucidate cases where the training distribution depends only on the class of the underlying labeling rule instead of the labeling rule itself. They then show that noisy parity and junta problems are PDS learnable by the gradient descent algorithm with a polynomial number of iterations. The authors further demonstrate the connection between PDS learnability and non-adaptive membership query learnability.

**Strengths:**

The paper is well-written and grounded in solid theory. The motivation behind the PDS framework is clearly presented: to illustrate the computational advantages of using an alternative feature distribution rather than the testing feature distribution. The authors use the PDS framework to demonstrate the learnability of several tasks via practical algorithms. These learnability problems are relevant and likely to interest conference attendees.

The DS-PAC learnability of the parity and $k$-junta problems by the gradient descent algorithm with ReLU networks is both interesting and novel. These findings not only show polynomial runtime and sample complexity for the parity and $k$-junta problems but also highlight the learning capability of practical deep learning algorithms.

**Weaknesses:**

The practicality of the PDS framework is unclear. For a practical learner, it is not feasible to choose or construct the training distribution based on the underlying labeling rule and/or the test distribution, since both are unknown to the learner. Consequently, DS learnability should be interpreted as an oracle model, where the training distribution is selected by an oracle. To address practicality without relying on an oracle, one option is to permit the learner to observe a few samples from the test distribution when constructing the training distribution. However, this setup may reduce to the non-adaptive membership query problem. Thus, the PDS framework may primarily serve as a theoretical tool for deriving non-adaptive membership query learnability, rather than as a practical method.

In particular, while the f-PDS setup is intriguing as a theoretical puzzle, its practical value is unclear. This definition assumes the training distribution is constructed adaptively with respect to the underlying labeling rule. In practice, a learner may need to observe labeled samples under the testing distribution to construct the training distribution. In such scenarios, it may be preferable for the learner to directly learn the classifier from the observed labeled samples under the testing distribution, rather than constructing a separate training distribution.

The statements of the theorems lack clarity. In the definitions of Defs 3.1, 4.1, and 4.2, learnability is defined as the existence of a well-behaved training distribution $D'$ for all testing distributions $D$. However, in Theorems 3.2, 4.3, and 4.6, for example, the authors also mention the existence of (another?) training distribution for all testing distributions, in addition to the learnability. This misalignment between the definitions and the statements of the theorems is confusing.

**Questions:**

- What concrete scenarios allow constructing a “well‑behaved” training distribution $D′$ without oracle access to the labeling rule or target distribution?
- Can the authors construct a practical procedure to synthesize $D'$?

---

> ### Author Response · Authors · 2025-11-21
>
> We thank the reviewer for their positive feedback and constructive comments.
>
> **Regarding Theorems 3.2, 4.3, 4.6**: Thanks for noticing this. Indeed, there is a redundancy in statements of Theorems 4.3 and 4.6 – the quantifier “there exists D’” should be removed from the theorem statement since this is already part of the definition. This is not a problem in Theorem 3.2, in which we do not mention D’ in the theorem statement. Is there a different issue in Theorem 3.2 that we are missing?
>
> **Regarding constructing the training distribution D’**:  The definition of DS-PAC allows D’ to be arbitrarily based on D, and we do not explicitly require it is easy to construct.  But in all our positive results, D’ is a mixture of a distribution D’’ that doesn’t depend on D, and D itself.  And so, it is easy to construct D’ based on access to a sampling oracle for D.
>
> More importantly: The way we view the realistic training procedure, which is in line with current practices, is **not that the learning rule constructs an arbitrary training distribution D’** (in the same way that a learner can choose query points in MQ or active learning).  Rather, part of the training procedure (as it actually happens in real life) is that we seek training data (which we can think of as coming from a training distribution) which is available to us, or that can be collected, and that is “good”.  This is currently done ad-hoc in a non principled way, and there has also been recent interest in empirical procedures for such training-set-selections (e.g. choosing the best training distributions in Pile, https://arxiv.org/abs/2403.16952, https://arxiv.org/abs/2407.01492, https://arxiv.org/abs/2305.10429, https://arxiv.org/abs/2507.09404).  Another example, in an even more complex domain, is https://arxiv.org/abs/2405.14838 where the training distribution involves a chain-of-thought, but this is “internalized” and prediction is made without using a chain-of-thought. We view our results, and more broadly the study of PDS which we are hoping to initiate here, as **giving principled guidance to “data collectors” in terms of what properties of a distribution (i.e. potential training data) would make it a good training distribution**.  We hope our work will give insight into what makes a good training distribution.  The main insight so far is that we should look for training distributions that more directly highlight correlations of intermediate computations and the target (individual features of the parity, or more broadly perhaps intermediate features).  This insight can be applied, e.g., to the internalized chain-of-thought example mentioned earlier, understanding it through a shifted distribution that provides more direct correlation to internal units of the final learned network.
>
> But we agree that **another important aspect is the theoretical connection to MQ**.  We would be very happy to see this re-ignite interest in MQ, and in particular non-adaptive MQ, which has not been studied as much on its own right (e.g. there are no lower bounds specifically on non-adaptive MQ, as far as we are aware).  Indeed, our view could help in driving new non-adaptive MQ methods (in fact, we have already made progress in this direction, studying sparse Fourier functions, which subsumes most prior work on non-adaptive MQ).  Moreover, we also want to draw attention to PDS specifically using SGD on a neural network: some MQ methods are based on things like comparing matched pairs of examples. This is different from SGD on a NN, and it would be interesting to understand if we could get the same benefit with this more benign and generic learning rule, putting the emphasis in the selection of the training distribution D’ (or query points) rather than a specialized algorithm.

---

> > ### Comment · Reviewer_9Uzc · 2025-11-26
> >
> > Thank you to the authors for their response.
> >
> > > More importantly: ...
> >
> > If I understand correctly, the existing theory states that under the uniform distribution, the best learning rule is computationally hard. Here, the best learning rule is defined as the best procedure that takes the training sample as input and outputs a classifier. Such procedures include those that internally discard some data, including the procedure the authors describe here. Therefore, the strategy of discarding some data should not work. When the learner is allowed to modify the data distribution using a small sample, the problem reduces to non-adaptive MQ.
> >
> >
> > In ICLR, the authors can upload a revised version. Your response would be clearer if you provided the revised paper.

---

### Meta-Review · Area_Chair_xeKu · 2026-01-08

**Summary:**

This submission suggests that distribution shift, which is often cast as an obstacle to learning, can help learning given carefully chosen distributions that train the learner. The paper shows how this so-called Positive Distribution Shift (PDS) appears in several forms, emphasizing that the benefit is often computational. Reviews contain a mix of positive and negative remarks. The theoretical direction is interesting and reviewers praised some of the results, while there are concerns about clarity, practical relevance, as well as notation (which to me is more minor than the review makes it out to be). From my view, indeed this is an interesting result and has technically correct support, but as the authors motivate in the clear introduction, they focus on a sort of transfer learning where one positively shifted training distribution can effect the learning advantageously. I would rephrase some of the concerns about practicality as much as scope, as the results, while rigorous and using ideas such as PAC learning, do not open many new theoretical avenues, and ultimately pertain to supporting a relatively narrow and important but unsurprising observation. There are several good ideas in this paper, but taken together, the reviews do not provide a consistently strong basis recommending acceptance.

**Reviewer Concerns:**

One primary outstanding concern is the limited practicality and scope of the method. For instance, reliance on oracle type information such as the dependence of the positive training distribution on the unknown target function, as well as somewhat limited empirical validation. There are some unresolved concerns about notation from one reviewer.

**Reviewer Scores:**

Scores are consistent with review content and reviewers participated in the rebuttal period

---

### Decision · Program_Chairs · 2026-01-26

Reject